# Nonparametric time series summary statistics for high-frequency accelerometry data from individuals with advanced dementia

**Keerati Suibkitwanchai**[1]*, **Adam M. Sykulski**[1], **Guillermo Perez Algorta**[2], **Daniel Waller**[1], **Catherine Walshe**[2]

**1** Department of Mathematics and Statistics, Lancaster University, Lancaster, United Kingdom, **2** Division of Health Research, Lancaster University, Lancaster, United Kingdom

* k.suibkitwanchai@lancaster.ac.uk

**Data Availability Statement:** All code and data are publicly available at https://doi.org/10.17635/lancaster/researchdata/386. All codes with R

## Abstract

Accelerometry data has been widely used to measure activity and the circadian rhythm of individuals across the health sciences, in particular with people with advanced dementia. Modern accelerometers can record continuous observations on a single individual for several days at a sampling frequency of the order of one hertz. Such rich and lengthy data sets provide new opportunities for statistical insight, but also pose challenges in selecting from a wide range of possible summary statistics, and how the calculation of such statistics should be optimally tuned and implemented. In this paper, we build on existing approaches, as well as propose new summary statistics, and detail how these should be implemented with high frequency accelerometry data. We test and validate our methods on an observed data set from 26 recordings from individuals with advanced dementia and 14 recordings from individuals without dementia. We study four metrics: Interdaily stability (IS), intradaily variability (IV), the scaling exponent from detrended fluctuation analysis (DFA), and a novel nonparametric estimator which we call the proportion of variance (PoV), which calculates the strength of the circadian rhythm using spectral density estimation. We perform a detailed analysis indicating how the time series should be optimally subsampled to calculate IV, and recommend a subsampling rate of approximately 5 minutes for the dataset that has been studied. In addition, we propose the use of the DFA scaling exponent separately for daytime and nighttime, to further separate effects between individuals. We compare the relationships between all these methods and show that they effectively capture different features of the time series.

## Introduction

Accelerometry data provides a method for monitoring physical activity over time. Such data is typically collected from an actigraph device, which is generally worn on the wrist, for a continuous period of time with minor impact on daily life. The device contains an accelerometer which measures acceleration of the individual, and sometimes a light and temperature sensor

Software for implementing our statistical methods are also publicly available at https://github.com/suibkitwanchai-k/Accelerometry.

**Funding:** This study was funded by the National Institute for Health Research (NIHR) HTA programme (project reference 15/10/11). The funders had no role in study design, data collection and analysis, decision to publish, or preparation of the manuscript.

**Competing interests:** The authors have declared that no competing interests exist.

is also included. Data recorded from such instruments has been widely used to study the pattern of 24-hour rest-activity rhythm, also known as circadian rhythm, of humans for almost half a century [1]. Of primary interest here is the increased use of actigraphy and accelerometry with people with advanced dementia at the end of life [2, 3]. However, the analysis of such data from this population where circadian rhythms are significantly dysregulated is challenging, putting into question the validity of conclusions extracted.

Several measures have been proposed to quantify the circadian rhythm and they can be broadly classified into two groups: parametric and nonparametric measures. Cosinor is a traditional parametric method for calculating the amplitude and phase of the circadian rhythm [4]. This procedure fits the observed data by a regression model of continuous cosine functions with a rhythm-adjusted mean, where the period is assumed to be known [5]. This model is often a poor fit to accelerometry data, especially for individuals with weak circadian patterns [6]. Alternative methods include singular spectrum analysis (SSA) which employs periodic components with varied amplitude and phase to fit the data [6], or the use of multiparameter-extended cosine functions [7]. In general, parametric methods such as cosinor can be very useful for characterising circadian rhythms, however such methods are typically model-based and suffer the usual disadvantages of parametric methods: namely the bias and lack of robustness that results when model assumptions are not met by the application data source of interest.

Nonparametric measures are model-free and require little to no user expertise to implement. Two such important and widely-used measures are interdaily stability (IS) and intradaily variability (IV). These were first proposed in 1990 to study how the circadian rhythm changes for patients with aging and Alzheimer's disease [8]. IS measures the strength of the circadian rhythm, while IV measures the *fragmentation* of the data by measuring the variation of hour-to-hour data relative to sample variance. In [8] and other such early studies, both IS and IV were calculated from data with an hourly sampling interval due to the limitations of sensor processing in the actigraph device. However, storage capacity has now been increased and data can be recorded with much higher temporal sampling intervals [9]. This is important as IV values change significantly as a function of the temporal sampling rate, although we note that the rate only has a very small effect on IS in general. In this paper we study the effect of temporal sampling on IV in detail, and propose an optimal rate, together with a simple method that ensures no data is thrown away in calculating the statistic.

IS and IV have been used as summary statistics for actigraphy and accelerometry data from individuals with dementia in several studies. A number of studies have reported that patients with dementia have significantly lower IS than those without dementia [8, 10–12], and this can be associated with cerebral microbleeds [13], occipital periventricular and frontal deep white matter hyperintensities [14], and loss of medial temporal lobe volume [15]. On the other hand, these patients have significantly higher IV [8, 10, 12, 16] which was positively correlated with the ratio of phosphorylated tau$_{81}$ (pTau) to cerebrospinal fluid amyloid β 42 (Aβ 42) for the biomarker collection assessing preclinical Alzheimer disease [17], and medial lobe atrophy [15]. More broadly, dementia can affect the disruption of circadian rhythm which is linked with dementia biomarkers [18].

Another nonparametric method is detrended fluctuation analysis (DFA), which was introduced to study fractal scaling behaviour and determine long-range autocorrelation in time series data [19]. With DFA, a scaling exponent is estimated, and this is related to the well-known Hurst exponent $H$ (proposed by Hurst [20]). The Hurst exponent value is between zero and one, and this can be divided into three cases: (1) $0.5 < H < 1$ for persistent time series whose increments have positive long-term autocorrelation, (2) $0 < H < 0.5$ for anti-persistent time series whose increments have negative long-term autocorrelation, and (3) $H = 0.5$ for a Brownian process with uncorrelated "white noise" increments. The DFA scaling exponent ($\alpha$),

on the other hand, is defined in the range $0 < \alpha < 2$ [21]. Stationary time series (such as a correlated noise) have $\alpha$ values between zero and one, whereas nonstationary time series (such as a random walk) have $\alpha$ values between one and two. The DFA scaling exponent is the same as the Hurst exponent in the stationary case such that $\alpha = H$, but in the nonstationary case we have $\alpha = H + 1$. The boundary between stationary and nonstationary time series behaves like "pink" noise (or 1/f noise) with $\alpha$ approximately equal to one. The DFA method has been commonly applied to time series with monofractal structure which is defined by a single scaling exponent. However, there might be some temporal fluctuations (multifractal structure) in the time series and this requires a group of generalised scaling exponents to explain them. Such fluctuations can be analysed by a method called multifractal detrended fluctuation analysis (MFDFA) [21, 22]. We note that there also exist other procedures for quantifying fractal-type behaviour in a time series (see e.g. [23]).

For its part, both DFA (or monofractal DFA) and MFDFA have been widely used to estimate scaling and Hurst exponents in many research fields. Indeed, a few research studies collected data from patients with dementia and analysed their fractal scaling behaviours using DFA [24–26]. Some of their findings were that the change of DFA scaling exponent could be found from ante-mortem actigraphy records of some patients with dementia and its degree of change was negatively correlated with the number of two major circadian neurotransmitters found in the suprachiasmatic nucleus [24], and the timed bright light therapy reduced the rate of decay of the Hurst exponent over time [25]. In this paper, we will further explore the use of monofractal DFA on accelerometry data, including a novel comparison of considering DFA scaling exponents during daytime and nighttime separately.

Finally, spectral analysis can be used to decompose variability in accelerometry data across different frequencies of oscillation. The simplest nonparametric estimator is the periodogram [27]. Some studies have preferred a modified version named the chi-squared periodogram [28] to characterise circadian rhythms [29–31]. For people who do not suffer from dementia or sleep disorders, both methods usually provide that the highest point in the spectrum generally occurs around the frequency of 1/24 hour$^{-1}$ (known as the fundamental frequency), and further peaks occur at *harmonic* frequencies, which are positive integer multiples of the fundamental frequency. Harmonic peaks occur as the circadian rhythm is not perfectly sinusoidal. For individuals with disrupted sleep cycles, the spectrum will have relatively low energy at the fundamental and harmonic frequencies, and relatively high and noisy levels at other frequencies. In this paper, we propose a novel nonparametric estimator from the periodogram which calculates the ratio of variability at fundamental and harmonic frequencies versus the whole spectrum. We call this method proportion of variance (PoV), and we will analyse this to our dataset and compare with other methods in order to assess the efficacy of spectral analysis methods for accelerometry data.

The main objective of this paper is to provide detailed methodology for calculating four different nonparametric time series summary statistics—IS, IV, the DFA scaling exponent, and PoV—to high frequency accelerometry data. The relationship among all statistical measures will be illustrated and analysed on a mixed dataset involving participants suffering from advanced dementia and participants who do not.

## Materials and methods

### Accelerometry data

The studied dataset contains data from 26 individuals suffering from advanced dementia. To create a reference group without dementia, we also studied 14 recordings of accelerometry data from members of the research team and colleagues that provided this data as part of

piloting the use of the actigraph device for our main study. The 26 individuals with advanced dementia took part in a group intervention project aimed at improving quality of life when living in care homes [32]. Fifteen of them received the Namaste Care intervention in their treatment during the period of data collection, while the remaining 11 participants were allocated to a non-intervention group receiving treatment as usual (see protocol details here [33]). Although the intervention did not have a significant effect on accelerometry readings of those with advanced dementia [32], we nonetheless compared intervention and non-intervention groups as a useful indicator of the performance and stability of our time series metrics. All the participants with advanced dementia were recruited from six nursing homes between the end of 2017 and May 2018. The study end date was 30/11/2018. Being a permanent resident in a nursing care home, lack of mental capacity, and a FAST score of 6-7 reflecting advanced dementia status were part of the inclusion criteria (see full details in [33]). All 26 participants with advanced dementia and with valid accelerometry information are part of a larger study detailed in [33], the demographic details of which are also included in S1 Table for reference.

The original trial was approved by the Wales Research Ethics Committee 5 Bangor Research Ethics Committee (reference number 17/WA/0378) on 22 November 2017. Potential participants were screened by the principal investigator and the senior care team, and eligible participant's written consent was provided by a personal consultee. For those without a personal consultee, the consent was requested to a nominated consultee following care home procedures. Following this, researchers discussed the study with consultees and gained assent from residents to take part in the study.

For each participant across the study, the accelerometry data was recorded by a wrist-worn accelerometer called GENEActiv for a maximum of 28 days. This device measures acceleration in the unit of gravitational force (g-force). The sampling period was set to five seconds. Raw data output from each sampling time consists of three non-negative values representing the magnitudes of projected vectors of acceleration on three axes in Euclidean space. The Euclidean norm was calculated from these magnitudes. Because the accelerometer is affected by the gravitation of Earth, this effect is removed. Thus, we calculate what is known as the *Euclidean norm minus one* (ENMO) [34], which results in the time series of accelerations used for implementing our nonparametric time series methods. We emphasise that in the literature other forms of actigraphy time series are sometimes studied [1] such as *time above threshold* (total amount of time such that the accelerometry data is above a setting threshold per time interval), *zero-crossing method* (total count of movements per time interval by counting the number of times when the accelerometry data is close to zero), and *digital integration* (total area under the curve of accelerometry data per time interval). We chose to run our analysis methods on ENMO data as this retained the most information from the raw data, but our methods could also have been applied to the types of actigraphy data described above. Before statistical analysis was performed, date ranges with missing data were excluded. We apply four different nonparametric time series methods (implemented in RStudio using R version 4.0.0) to the ENMO data which we now describe. R Software for implementing each method, for any generic accelerometry time series, is publicly available at www.github.com/suibkitwanchai-k/Accelerometry.

## Interdaily stability (IS)

The first statistical method is interdaily stability (IS), which is a nonparametric measure for the strength of the circadian rhythm in the accelerometry data. The circadian rhythm is driven by a circadian clock with approximately 24-hour time period. For each individual, we define $\{X_t\}$ as a time series of ENMO data with length $N$. The interdaily stability is the ratio of the average

square-error of hourly means from the overall mean to the variance of $\{X_t\}$, i.e.

$$
\text{IS} = \frac{\sum_{s=1}^{24}(\overline{X}_s - \overline{\overline{X}})^2/24}{\sum_{t=1}^{N}(X_t - \overline{\overline{X}})^2/N},
\tag{1}
$$

where $\overline{\overline{X}}_s$ is the sample mean of ENMO data during the $s^{\text{th}}$ hour of the day ($s = 1, 2, \ldots, 24$) and $\overline{\overline{X}}$ is the overall sample mean. Eq (1) calculates IS by aggregating data on an hourly basis in the numerator into 24 bins. The original motivation for this is that the time series itself is sampled on an hourly interval [8] such that more bins are not possible. However for high frequency data, such as the 5-second data in our study, the hourly aggregation becomes an arbitrary choice, and IS could instead be calculated by aggregating over shorter time intervals and into several more bins. According to [9] however, the interdaily stability is insensitive to this choice, and we found the same in our study. Therefore we keep to the choice of hourly binning in Eq (1). In contrast, the value of intradaily variability (IV) depends strongly on the sampling period as we now discuss.

## Intradaily variability (IV)

Intradaily variability (IV) is used to estimate circadian fragmentation in the data. In our implementation, we ensure all data in the high frequency time series is used, even if subsampling is applied. Specifically, we let

$$
\{Y_k^{[j]}\} = \{X_{(k-1)\Delta+j, k=1,2,\ldots,\lfloor N/\Delta \rfloor}\}
\tag{2}
$$

be a finite sequence obtained from subsampling the time series of ENMO data $\{X_t\}$ with length $N$ by an arbitrary positive integer $\Delta$ not greater than $N$, and $j$ is the index of the $j^{\text{th}}$ time series data derived from this subsampling ($1 \leq j \leq \Delta$). For each $\{Y_k^{[j]}\}$, the intradaily variability is calculated by the ratio of variation in consecutive time intervals to the total variance, i.e.

$$
\text{IV}[j] = \frac{\sum_{k=2}^{M}(Y_k^{[j]} - Y_{k-1}^{[j]})^2/(M-1)}{\sum_{k=1}^{M}(Y_k^{[j]} - \overline{Y^{[j]}})^2/M},
\tag{3}
$$

where $M = \lfloor N/\Delta \rfloor$ is the length of $\{Y_k^{[j]}\}$ and $\overline{Y^{[j]}}$ is its overall mean. The final IV value is then the average of all IV values obtained from Eq (3), i.e.

$$
\text{IV} = \frac{\sum_{j=1}^{\Delta}\text{IV}[j]}{\Delta}.
\tag{4}
$$

This implementation, which averages across different start points of the time series, ensures all data is used regardless of the choice of $\Delta$. Note that this procedure was also proposed in [35] for calculating integrated volatility in financial time series, which is a very similar metric to IV.

In [8] the subsampling period was set to 60 minutes. However, for higher frequency time series, it can be adjusted to any other appropriate time [6, 9]. In the next section we shall investigate in more detail the relationship between IV and the choice of $\Delta$ for our studied dataset. We shall show that there is a natural trade-off to be balanced in that overly low values for $\Delta$

lead to poor estimates due to short-lag autocorrelations, whereas large values for $\Delta$ fail to capture the differences in fragmentation for individuals with and without advanced dementia. Another consideration we make is the correlation between IV and other statistical measures, and whether this metric responds as it should to other summary statistics for a given subsampling period.

## DFA scaling exponent

The Hurst exponent ($H$) ranges between zero and one, and it is used to measure the degree of long-range dependence of a time series, which is the rate of decay of its long-lag autocorrelation. A higher Hurst exponent indicates that the time series data is more persistent and has higher degree of positive long-term autocorrelation. Detrended fluctuation analysis (DFA) can be used to estimate $H$ via the DFA scaling exponent ($\alpha$) which ranges between zero and two such that $\alpha = H$ for stationary noise-like behaviour and $\alpha = H + 1$ for nonstationary random walk-like behaviour. Here we succinctly summarise the estimation of the DFA scaling exponent for our data in the following steps (see also [21]):

1. For a time series of ENMO data $\{X_t\}$ with length $N$, we construct a new time series $\{Z_t\}$ which is the cumulative sum of $\{X_t\}$ with mean centering, i.e.

$$Z_t = \sum_{u=1}^{t}(X_u - \overline{X}),\tag{5}$$

   where $1 \leq t \leq N$.

2. We divide $\{Z_t\}$ into equally-sized non-overlapping segments with length $S$ (scale parameter) where $S$ is chosen to vary across a range. According to [21], a minimum size of $S$ of larger than 10 is considered to be a rule of thumb. For our accelerometry data, we set $S = 2^i$, where $i$ is varied from 4 to 8 in increments of 0.25. Since our sampling period is 5 seconds, this range of $S$ is therefore between 1.333 and 21.333 minutes. Larger values of $S$ could have been applied to our data but we found this makes the method more computationally expensive without significantly changing the estimated exponent. A least squares regression line is then fitted to the time series data in each segment for each value of $S$. The local root-mean-square deviation at segment number $K$ with length $S$ is calculated as

$$\text{RMSD}_{S,K} = \sqrt{\frac{\sum_{j=S(K-1)+1}^{SK}(\hat{Z}_j - Z_j)^2}{S}},\tag{6}$$

   where $\hat{Z}_j$ is the predicted value from the regression line and $1 \leq K \leq \lfloor N/S \rfloor$.

3. For each $S$, the overall root-mean-square deviation ($F$) is calculated as

$$F(S) = \sqrt{\frac{\sum_{i=1}^{\lfloor N/S \rfloor}\text{RMSD}_{S,i}^2}{\lfloor N/S \rfloor}}.\tag{7}$$

4. The DFA scaling exponent ($\alpha$) is estimated by the slope of the linear regression line between $\log_2(S)$ (x-axis) and $\log_2(F)$ (y-axis).

In our analysis, the DFA scaling exponent is estimated individually for each participant across the whole time series. We also calculate separate values for each participant from daytime and nighttime readings. This is done to examine whether there is a difference between daytime and nighttime activity for any given participant, and to see whether this difference is more pronounced in any of the participant groups, thus providing more statistical insight from this type of analysis.

## Proportion of variance (PoV)

The power spectral density (PSD) or power spectrum describes the distribution of variability in a time series as a function of frequency. It is equivalent to Fourier transform of the autocovariance sequence of a stationary time series. The periodogram is an asymptotically unbiased estimate of the PSD which is defined by

$$I(f) = \frac{1}{N} \left| \sum_{t=1}^{N} (X_t - \overline{X}) e^{-i2\pi ft} \right|^2, \tag{8}$$

where $N$ is the length of ENMO time series $\{X_t\}$, $\overline{X}$ is its overall mean, and $f$ is the frequency in cycles per second or hertz (Hz). $i \equiv \sqrt{-1}$ represents the imaginary unit and $|\cdot|$ denotes the absolute value. For $\{X_t\}$ with real-valued data, the periodogram is real-valued and symmetric, i.e. $I(f) = I(-f)$. In addition, as $\{X_t\}$ is a discretely observed time series from sampling, then its periodogram is only defined for positive and negative frequencies that are less than half of the sampling rate, which is also called the Nyquist frequency ($f_n$). For our data, which was sampled every 5 seconds (sampling rate = 1/5 Hz), the Nyquist frequency is therefore equal to $f_n = 1/10$ Hz. The total area under the spectral line of the periodogram between $-f_n$ and $f_n$ is approximately equal to the variance of $\{X_t\}$.

The periodogram can be used to assess the strength of circadian rhythm by evaluating its spectral value at the frequency $f = 1/24$ hour$^{-1}$ (1/86400 Hz). However, because the total variance of ENMO data will vary across individuals, directly comparing the spectral values of the periodogram at just one frequency will be a noisy and unreliable measure of circadian strength. Instead, we propose a new method by calculating the proportion of variance (PoV) explained by the periodogram at or *near* the fundamental frequency (also called the first harmonic) of circadian rhythm ($f = 1/24$ hour$^{-1}$). Furthermore, because circadian patterns will typically not be perfect sinusoids, we also include variability from the three following harmonic frequencies. The PoV statistic is then obtained by finding the ratio of area under the spectral line around the frequencies of interest to the total sample variance. Specifically, we use the range of frequencies between $|f| = 1/24.5$ hour$^{-1}$ (1/88200 Hz) and $|f| = 1/23.5$ hour$^{-1}$ (1/84600 Hz) along with their positive integer multiples from two to four. In the simplest case of considering only the fundamental frequency and ignoring higher orders of harmonics, the PoV value for each individual is calculated as

$$\mathrm{PoV}^{(F)} = \frac{\displaystyle\int_{-1/84600}^{-1/88200} I(f)df + \int_{1/88200}^{1/84600} I(f)df}{\mathrm{Var}(X_t)} = \frac{2\displaystyle\int_{1/88200}^{1/84600} I(f)df}{\mathrm{Var}(X_t)}, \tag{9}$$

where $f$ is the frequency in hertz and $\mathrm{Var}(X_t) = \sum_{t=1}^{N} (X_t - \overline{X})^2/(N-1)$. The values from negative frequencies are included in the numerator in Eq (9) as these contribute to the total variance. Due to the symmetry of the periodogram, this contribution is equivalent to the corresponding contribution from positive frequencies which yields the simpler final expression.

Therefore PoV$^{(F)}$ can be interpreted as the proportion of variance in the time series explained by absolute frequencies between 1/24.5 hour$^{-1}$ and 1/23.5 hour$^{-1}$. In the case of considering the first four harmonic frequencies, the PoV value for each individual is calculated as

$$\text{PoV}^{(H)} = \frac{\sum_{k=1}^{4} \int_{-k/84600}^{-k/88200} I(f)df + \sum_{k=1}^{4} \int_{k/88200}^{k/84600} I(f)df}{\text{Var}(X_t)} = \frac{2\sum_{k=1}^{4} \int_{k/88200}^{k/84600} I(f)df}{\text{Var}(X_t)}, \tag{10}$$

where again $f$ is the frequency in hertz and the steps in the derivation follow the same pattern as Eq (9). The number of harmonics used can be amended if necessary, but we found 4 to be a good value to use in practice.

There are several alternatives to using the periodogram, including smoothed and multi-tapered approaches. However, these techniques smooth across frequency leading to a loss in data resolution. Since our statistics in Eqs (9) and (10) are already implicitly smoothing across frequencies near the peak frequency (and harmonics) to account for noise and variability, then we found no such further smoothing is required.

## Results and discussion

### Accelerometry data

In the left column of Fig 1, we display three examples of time series of ENMO data, one from each of the different groups (non-intervention, intervention, and without dementia). The ENMO values from individuals with advanced dementia (non-intervention and intervention groups) were largely between 0 and 0.2g with occasional spikes above this range, and this was consistent with other participants in the study from these groups. The data from the individual without dementia had higher ENMO values and its daily circadian pattern was more clearly observed. In the right column of Fig 1, we display the 24-hour time average plots of ENMO data from these three individuals. The average data from the individual without dementia had high ENMO values with large fluctuation during daytime and low ENMO values with moderate fluctuation during nighttime. The average data from individuals with advanced dementia, however, had low ENMO values throughout the day and no clear circadian cycle was present.

### Interdaily stability (IS)

We used IS to evaluate the strength of the circadian rhythm in the ENMO data. The bar chart in Fig 2 displays IS values from all participants in the study. All participants without dementia except the 7th and 14th recordings had higher IS values than all participants with advanced dementia. The corresponding box plot in Fig 2 shows that participants in the intervention group had slightly higher mean IS value than in the non-intervention group, but this was not found to be significantly different (p-value = 0.413). However, the mean IS value from the group without dementia was significantly higher than from the combined non-intervention and intervention group (p-value <0.001). All p-values in this paper were calculated using the nonparametric Mann-Whitney U-test at a 5% significance level. S2 Table in the supporting information presents mean, standard deviation, minimum and maximum values of all our metrics across all participant groups.

### Intradaily variability (IV)

We used IV to assess the circadian fragmentation in the ENMO data. First we attempted to find the appropriate subsampling interval with which to calculate IV in Eqs (2)–(4). To do this,

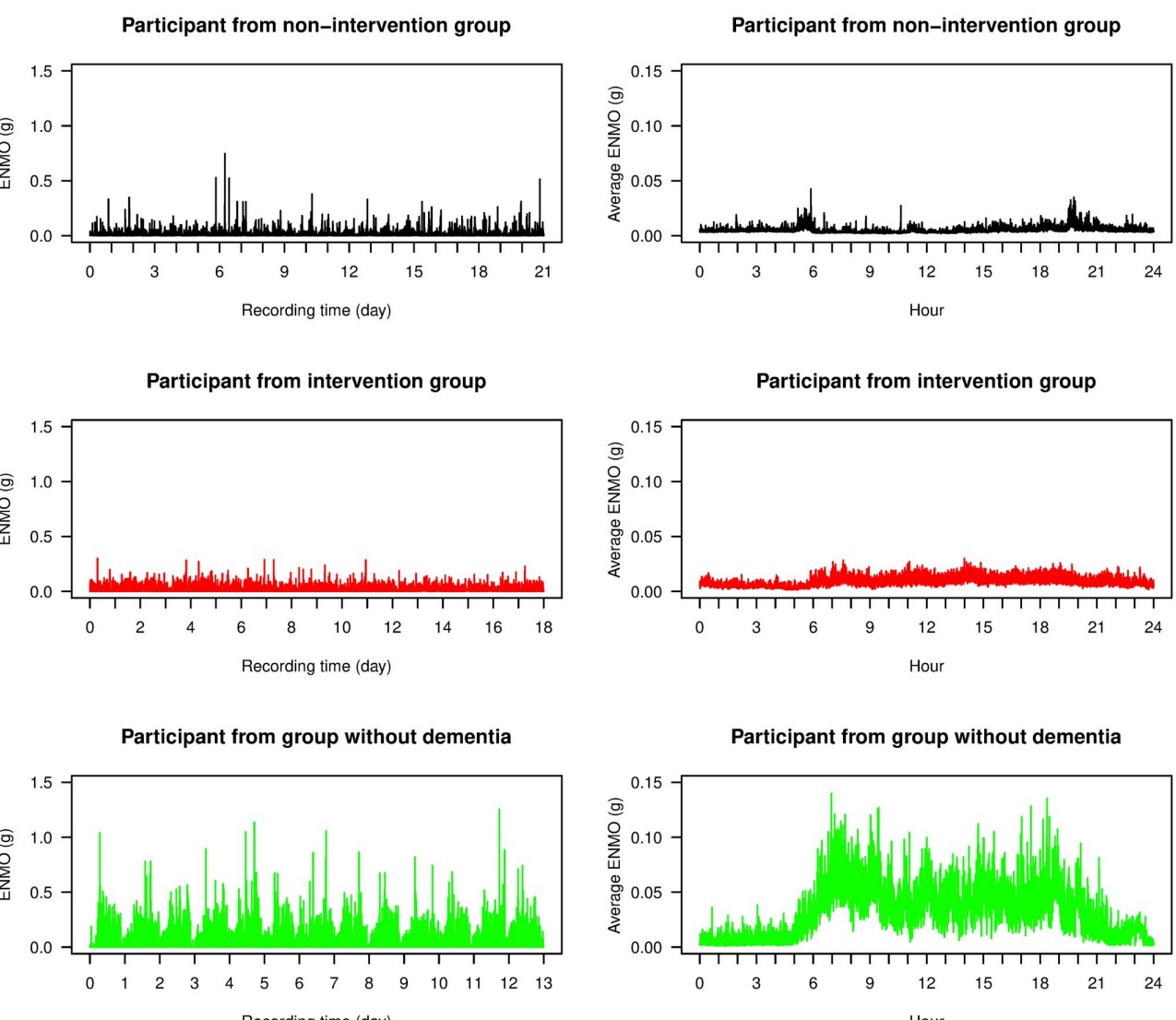

**Fig 1. ENMO time series.** Time series plots of ENMO data (left) and their 24-hour time average plots (right) from three participants from the study: one from each of the non-intervention group (black), the intervention group (red), and the group of individuals without dementia (green).

we subsampled the ENMO data with a sampling period ranging from 5 seconds ($\Delta = 1$) to 60 minutes ($\Delta = 720$), in intervals of 5 seconds, and calculated IV with each sampling period. Due to the high sampling frequency of our data, the conventional subsampling interval, which is hourly subsampling, was not found to be appropriate. The left plot of Fig 3 shows the relationship between the average IV value and the subsampling interval for the three different groups of individuals. IV values increased as a function of subsampling interval for each group, converging to one another as the subsampling interval approached one hour. This pattern of IV as a function of subsampling interval is generally consistent with Figs 2, 3 and 5 of [9], but other more "U-shaped" relationships are found in Fig 4 of [9] and Fig 6 of [6]. We note however that these studies were on a range of simulated data, as well as human and animal participants, and used different actigraphy output by studying the counts per time interval rather than ENMO accelerometry data. Mathematically though, both types of observed relationships between subsampling interval and IV are possible, and the shape of this relationship will depend on the

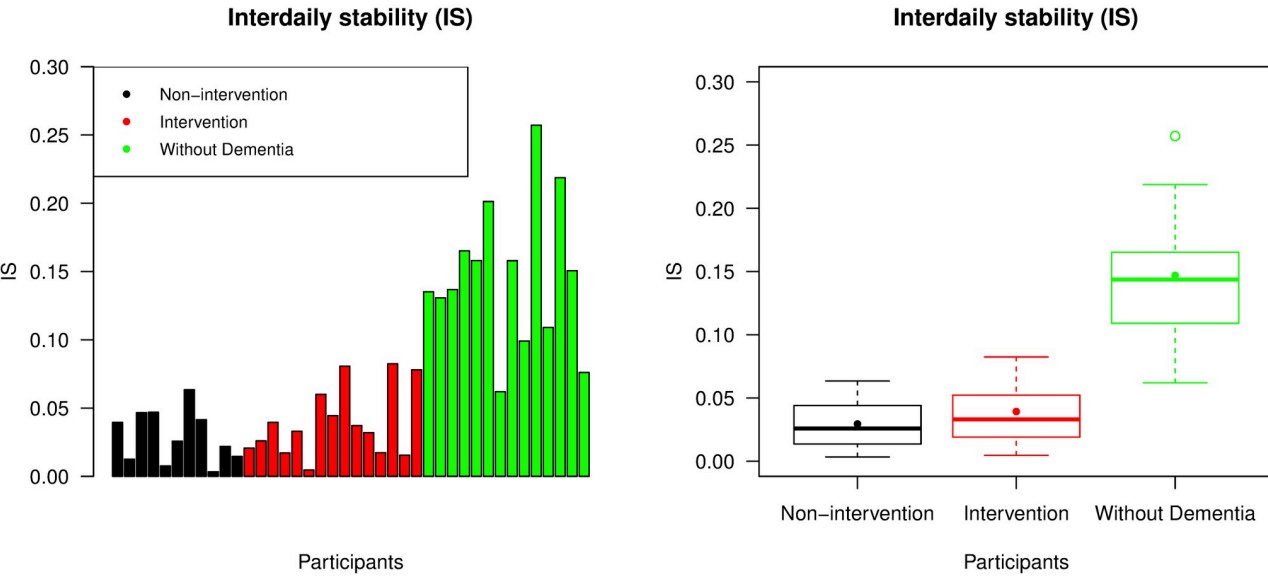

**Fig 2. Interdaily stability (IS).** The left panel displays IS values across all participants and the right panel is a box plot collating these values across the non-intervention group, the intervention group, and the group of individuals without dementia. For each group, the median and mean of the IS values are presented by a thick line and a point in its own box, respectively.

autocorrelation characteristics of the data, as well as the recording device used and the type of accelerometry output studied. In our case, the initial rapid rise in IV as the subsampling interval was increased from 5 seconds to higher multiple values was due to the large positive autocorrelations in the time series at these small lags. In the right plot of Fig 3, we display the same results as the left plot, but this time the IV values from the intervention group and the group of individuals without dementia are represented as percentages of the values from the non-intervention group across the subsampling interval. Here we see that lower subsampling intervals

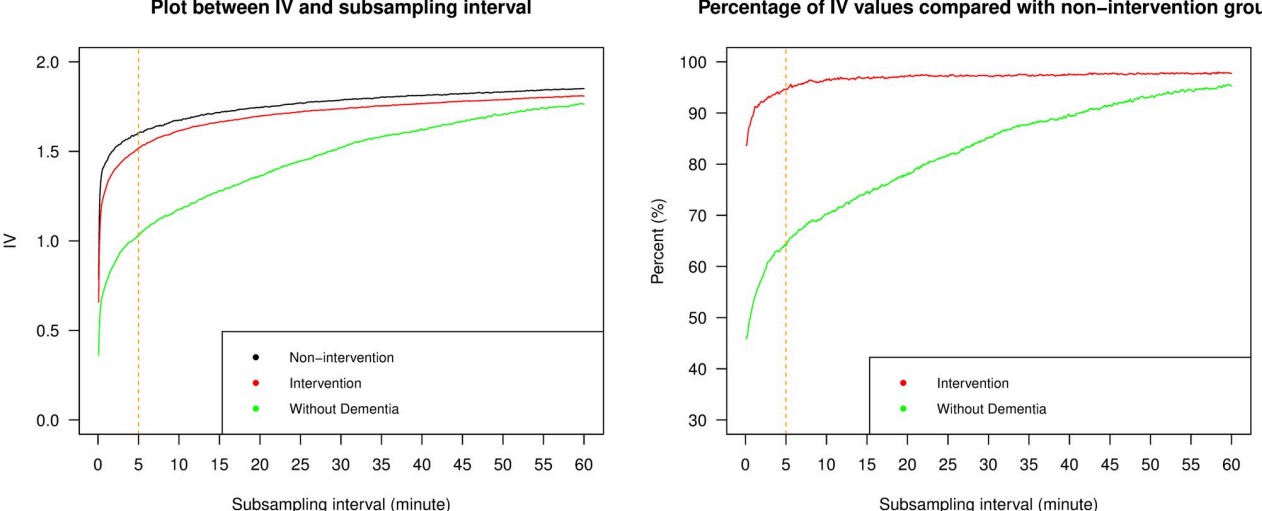

**Fig 3. Intradaily variability (IV) vs. subsampling interval.** The left panel displays line plots of average IV values against the subsampling interval for the three different groups of individuals; the right panel displays line plots of the percentage of average IV values from the intervention group and the group of individuals without dementia, compared with those from the non-intervention group. The orange dashed line refers to 5 minutes and indicates our recommended subsampling interval.

were more effective at separating the groups. This indicates a clear trade-off in selecting the optimal subsampling interval.

To further study the appropriate choice of subsampling interval for IV, we investigated the effect of subsampling on the relationship between IV and our three other statistical measures. In the supporting information, we show plots of the Pearson's correlation coefficients, which were used to determine the linear association between IV and the other measures (S1–S3 Figs). Because IV is used to assess the circadian fragmentation, then the measurement of strength of the circadian rhythm, as measured by IS or PoV, should have an associated negative correlation. Similarly, there should be a negative correlation between IV and the DFA scaling exponent, as high IV and low DFA exponents are synonymous with rougher more volatile time series, whereas low IV and high DFA exponents correspond to smoother more meandering time series. The results in the supporting figures indeed revealed such negative correlations between IV and three other metrics, but only within suitable ranges of subsampling intervals that were not too high or too low, indicating that IV performed as expected in this range.

Taking together the evidence from Fig 3 and S1–S3 Figs, we suggest selecting a subsampling interval in the range from 2 to 10 minutes. We selected 5 minutes when reporting our results that follow, as indicated by the dashed line in Fig 3 (although findings were broadly consistent selecting any value in this range). The range of 2 to 10 minutes partially overlaps with the findings of [9] where the most significant difference between groups of participants with and without dementia occurred when the subsampling rate was between 5 and 45 minutes. As mentioned, discrepancies between findings are likely due to the type of study being performed, as well as the specific recording device used, and we recommend such analyses are repeated in future accelerometry studies to ascertain the most appropriate subsampling rate which is an important choice for IV to perform meaningfully as a summary statistic.

Using a subsampling interval of 5 minutes, the corresponding IV value was calculated for each individual, as presented in Fig 4. Participants from both non-intervention and intervention groups usually had higher IV values than participants without dementia, as expected. As was the case with IS values, the participants without dementia had a larger spread of IV values than participants with advanced dementia. This was likely due to participants without dementia not having similar living conditions to each other, unlike participants with dementia who all resided in care homes. The mean IV value from the group of individuals without dementia was less than that from the combined non-intervention and intervention group (p-value <0.001), but there was no significant difference in the mean IV value between these two groups of individuals with advanced dementia (p-value = 0.357).

## DFA scaling exponent

The DFA scaling exponent ($\alpha$) from each individual was estimated by the slope of the linear regression line between scale and overall root-mean-square deviation in log-log coordinates, as described in the Materials and Methods section. In Fig 5, the regression lines from three individuals from the different groups are presented. We evaluated the coefficient of determination ($R^2$) as a measurement of goodness of fit and found that approximately 99% of the variability of the data can be explained by the linear regression model in each case. From this figure, we can see that the participant without dementia has a higher slope, and hence higher $\alpha$ value, than the participants from the non-intervention and intervention groups. We found this figure to be generally representative of the differences between DFA scaling exponents across all participant groups.

Fig 6 shows DFA scaling exponents from all participants. The $\alpha$ values from the group of individuals without dementia were likely to be higher than those from the other groups (p-

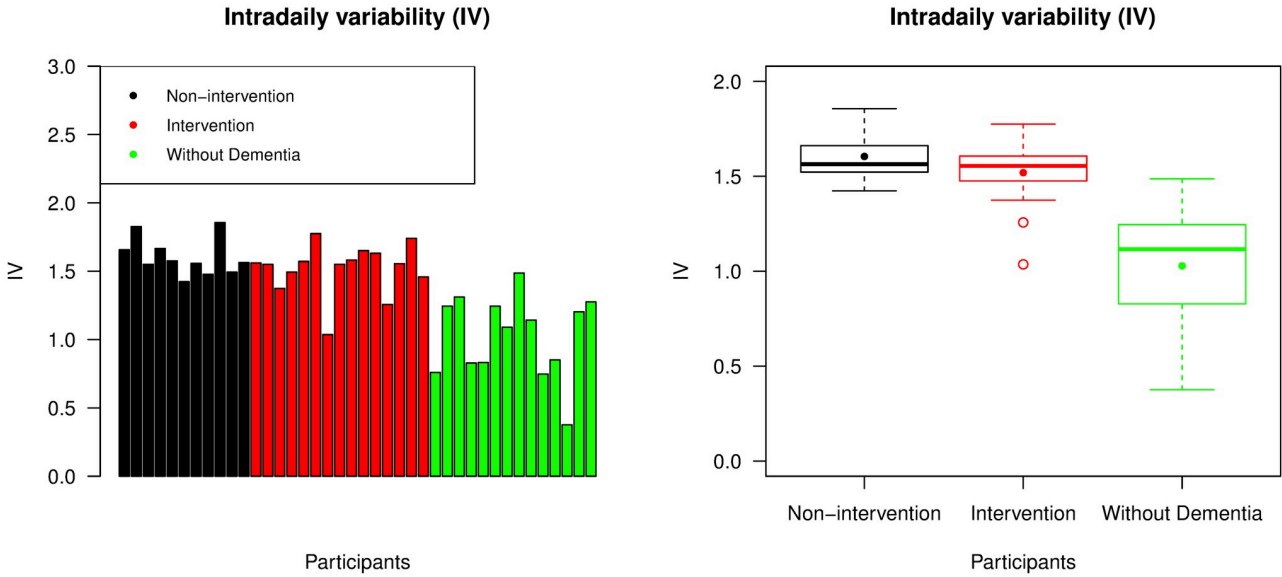

**Fig 4. Intradaily variability (IV).** The left panel displays IV values across all participants, with the same ordering of individuals as the bar chart representing IS values; the right panel is a box plot collating IV values across the non-intervention group, the intervention group, and the group of individuals without dementia. For each group, the median and mean of the IV values are presented by a thick line and a point in its own box, respectively.

value $<0.001$). In addition, participants from the intervention group had significantly higher mean $\alpha$ value than the non-intervention group (p-value = 0.004). All $\alpha$ values from these two groups with advanced dementia were between 0.7 and 1. This suggests that their corresponding ENMO time series were stationary and noise-like but with long-term positive autocorrelations. For participants without dementia, $\alpha$ values were found to be very close to 1. The time

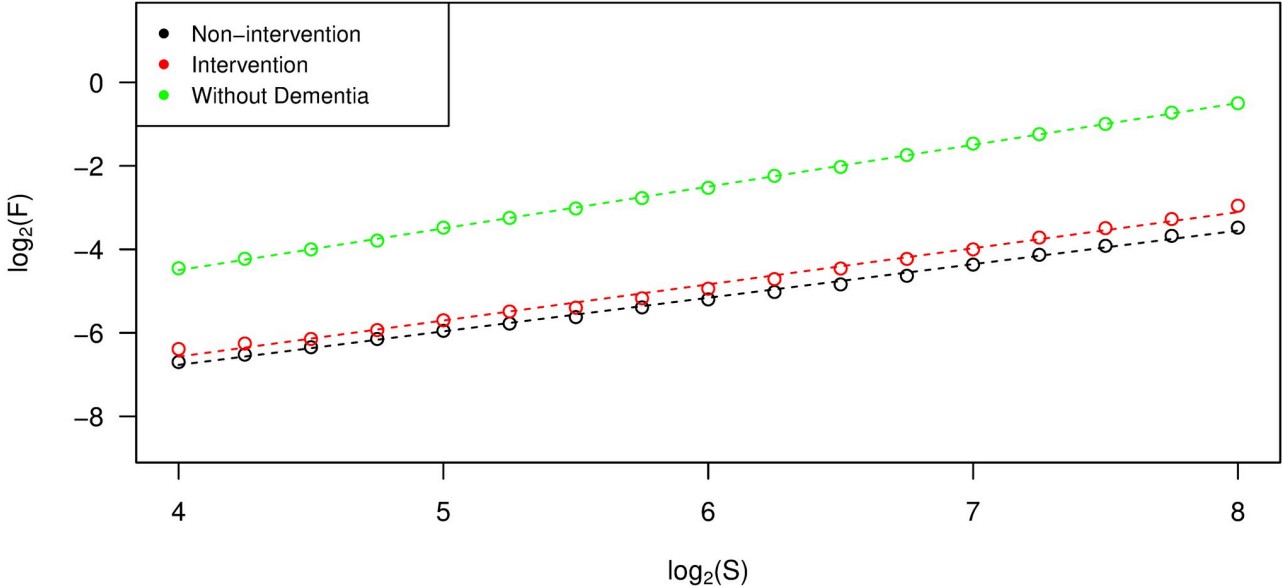

**Fig 5. Estimation of DFA scaling exponents.** Data points and their linear regression lines for the estimation of DFA scaling exponents (slopes) from three participants, one from each of the non-intervention group, the intervention group, and the group of individuals without dementia.

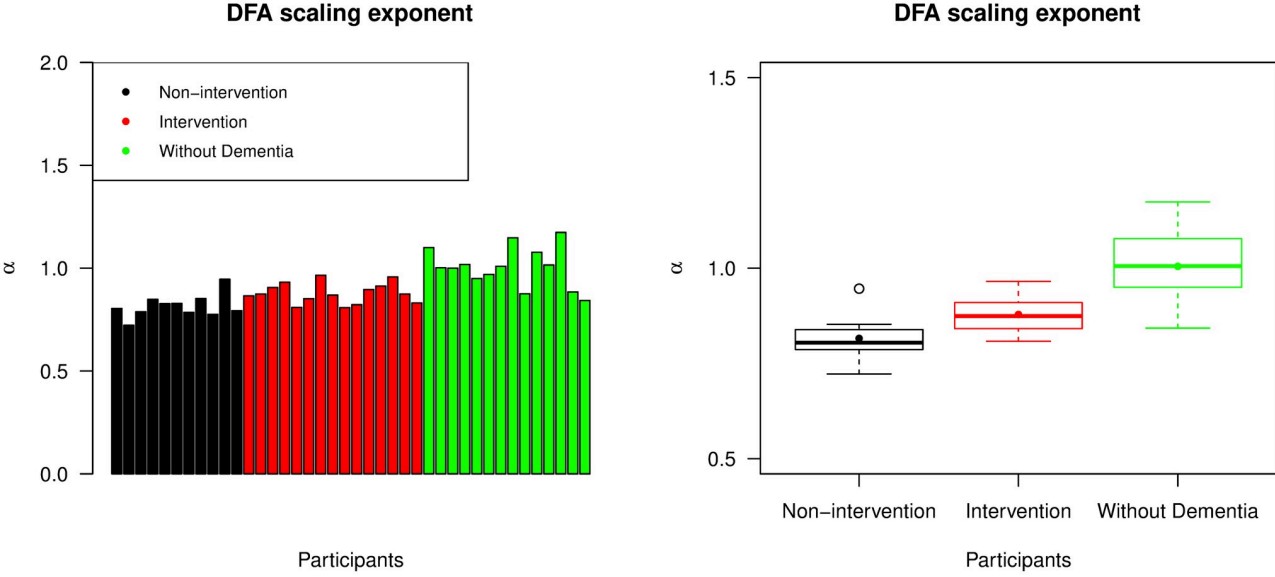

**Fig 6. DFA scaling exponents ($\alpha$).** The left panel displays $\alpha$ values across all participants, with the same ordering of individuals as the bar chart representing IS values; the right panel is a box plot collating $\alpha$ values across the non-intervention group, the intervention group, and the group of individuals without dementia. For each group, the median and mean of the $\alpha$ values are presented by a thick line and a point in its own box, respectively.

series with these $\alpha$ values were in the boundary between stationary noise and nonstationary random walks and it was difficult to classify into either group. This type of time series data is sometimes called pink noise or 1/f noise. In general, the higher values of $\alpha$ observed in the group of individuals without dementia are consistent with time series that exhibited smoother and less jittery trajectories.

We also calculated separate DFA scaling exponents from daytime and nighttime readings of the ENMO data. Because each participant had a different sleeping period, and there was no available data from the care homes regarding daily schedules, then setting fixed daytime and nighttime periods *a priori* was challenging. Instead we took a data-driven approach. Fig 7 displays the average EMMO readings throughout the day, averaged across each participant group and across all participants. In general, we observed low ENMO values between 11 p.m. and 6 a.m. and as such we used this seven hour interval as *nighttime* readings, and ENMO values between 6 a.m. and 11 p.m. as *daytime* readings.

Fig 8 shows two box plots presenting the summary statistics of separate DFA scaling exponents from daytime and nighttime readings. The daytime readings had a similar distribution of $\alpha$ values to the entire time series analysed in Fig 6. The nighttime readings however provided a rather different distribution. Their $\alpha$ values were more similar across the groups and indeed there was no longer a significant difference of mean $\alpha$ value between participants in the intervention group and the group of individuals without dementia (p-value = 0.747). For the non-intervention and intervention group, there was no significant difference of mean $\alpha$ value between daytime and nighttime readings (p-values = 0.949 and 0.305). Participants without dementia, by contrast, had a significant difference of mean $\alpha$ value, with daytime values being higher (p-value = 0.008). These results suggested that participants with advanced dementia tended to have similar records of accelerometry throughout the day, and no significant change of fractal behaviour of data between daytime and nighttime. However, participants without

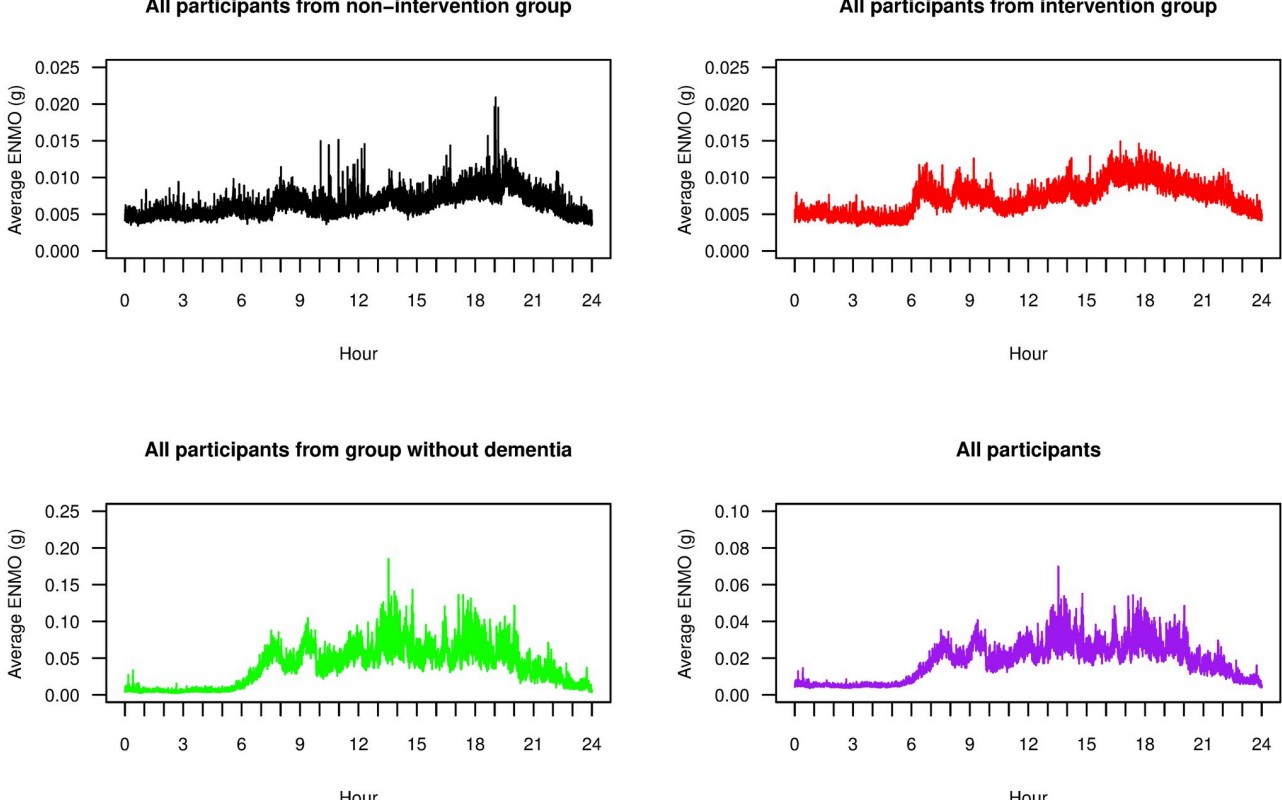

**Fig 7. Average ENMO over participants.** Time series plots of 24-hour ENMO values averaging over all participants in each of the non-intervention group (black), the intervention group (red), the group of individuals without dementia (green), and all groups together (purple).

dementia were likely to have this change of behaviour such that their accelerometry data could be separately described by two fractal exponents, one for daytime and one for nighttime.

## Proportion of variance (PoV)

In Fig 9, we display the power spectral density, as estimated by the periodogram, from one participant from each of the non-intervention group, the intervention group, and the group of individuals without dementia, along with their corresponding percentage of the total variance at each frequency. The periodogram for the participant without dementia had its highest peak at a frequency of 1/24 hour$^{-1}$, which we call the first harmonic or fundamental frequency, and the next highest peaks were at the following higher orders of harmonics. These features were found in all other participants without dementia. Thus, their ENMO data was roughly periodic with a time period of approximately 24 hours or one day, corresponding to the time period of the circadian rhythm. However, accelerometry data from individuals with advanced dementia was often distorted and this resulted in a much less clear pattern of circadian rhythm. In particular, the periodograms showed that the ENMO time series were less periodic, and the time period of the circadian rhythm was not well represented by the periodogram. In Fig 9, the periodogram for the participant in the non-intervention group did not show a clear and sharp peak at some of the harmonics including the fundamental frequency. Although the highest peak could be observed at the fundamental frequency for the participant in the intervention group, there were less evident peaks at higher orders of harmonics and some peaks are located outside these frequencies. These characteristics were generally representative of the

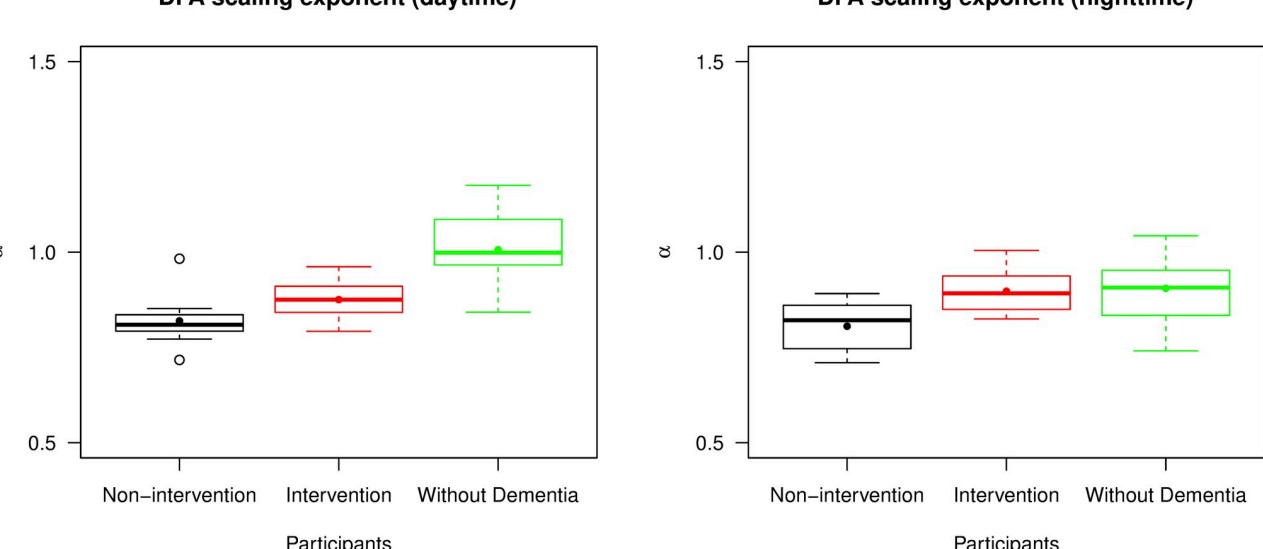

**Fig 8. Distribution of DFA scaling exponents ($\alpha$) from daytime and nighttime readings.** The left panel is a box plot collating $\alpha$ values from daytime readings and the right panel is a box plot collating $\alpha$ values from nighttime readings across the non-intervention group, the intervention group, and the group of individuals without dementia. For each group and type of readings, the median and mean of the $\alpha$ values are presented by a thick line and a point in its own box, respectively.

participants from each respective group, however we note that there was variability across participants with advanced dementia in terms of the location of dominant peaks in the spectrum —the idea behind our PoV statistic is to smooth over such noisy characteristics to obtain a stable metric.

First we used Eq (9) to calculate PoV$^{(F)}$, which is the ratio of area under the spectral line around the fundamental frequency to the total variance of the time series (multiplied by 2 to include the negative frequency). This was used to indicate how well the variance was captured by periodic waves with period of 23.5-24.5 hrs. The plots in Fig 10 indicate that participants without dementia had significantly higher PoV$^{(F)}$ values than participants from the other two groups (p-value <0.001). Overall, we see that this metric explained on average 15.161% of the variance for participants without dementia but typically less than 10% for participants with advanced dementia. We note that there was no significant difference in the PoV$^{(F)}$ values between intervention and non-intervention groups (p-value = 0.217).

To try and explain more of the variability, we then calculated PoV$^{(H)}$ from Eq (10) which included all periodogram values around the first four harmonic frequencies. The results are presented in Fig 11. Participants without dementia again had significantly higher PoV values than those with advanced dementia (p-value <0.001), whereas there was no significant difference between intervention and non-intervention groups (p-value = 0.384). More variability was captured by this metric, in particular in the group of individuals without dementia where an average of 22.418% of the variance was captured. This shows the benefits of including harmonic frequencies.

Our proposed method conceptually shares some similarities with the well-known cosinor method in that both methods measure the strength of periodic waves existing in the data. Indeed, we found that the coefficient of determination of the cosinor model (with 24-hour time period) was very highly positively correlated with PoV$^{(F)}$ ($\rho$ = 0.977) across all participants. However, when we included harmonics in PoV, then this correlation was still strong

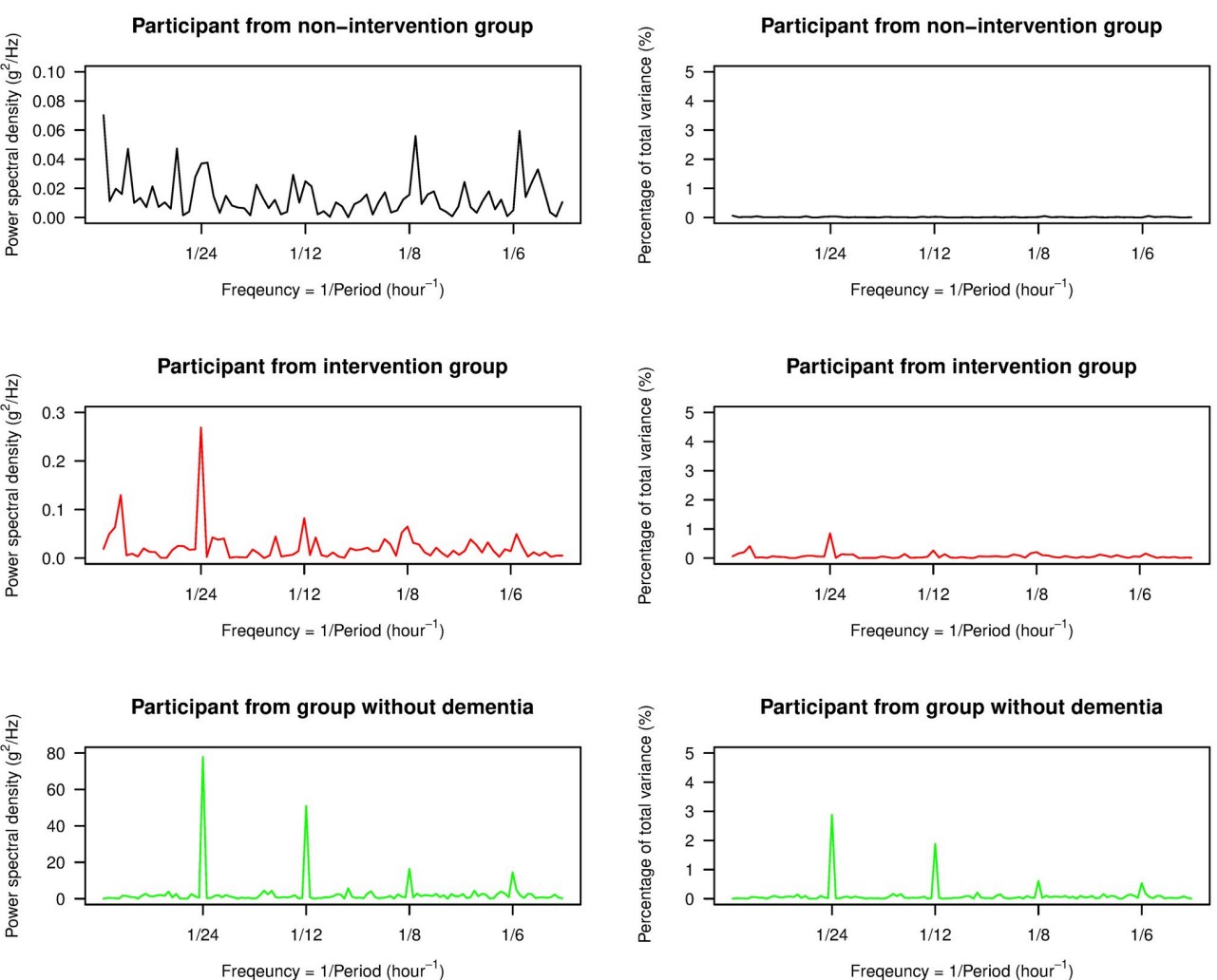

**Fig 9. Periodograms.** The left panel displays the plots of the power spectral density estimated by the periodogram for three participants, one from each of the non-intervention group, the intervention group, and the group of individuals without dementia. The right panel displays the plots of the percentage of the total variance captured at each frequency. The range of frequencies for each plot in both panels includes all four harmonic frequencies used in our study.

but dropped to 0.901. Overall, we found the inclusion of harmonics to better represent the data and better separate behaviour across groups, and this is because of the non-sinusoidal nature of the circadian cycle which $PoV^{(H)}$ could capture, but $PoV^{(F)}$ and the cosinor method could not. As evidence of this, $PoV^{(H)}$ captured an average of 22.418% of the variability in participants without dementia, whereas with the cosinor method only 7.938% was captured.

## Comparison of statistical measures

This section explores the relationships of results from all four proposed nonparametric methods. Scatter plots between all possible pairs of the four statistical measures—IS, IV (using a subsampling interval of 5 minutes), the DFA scaling exponent or $\alpha$ (from the whole range of the data), and $PoV^{(H)}$—are shown in Fig 12. We see that the statistics of participants from intervention and non-intervention groups were largely non-separable by any pair of these measures. However, the statistics of participants without dementia were mostly distinct from

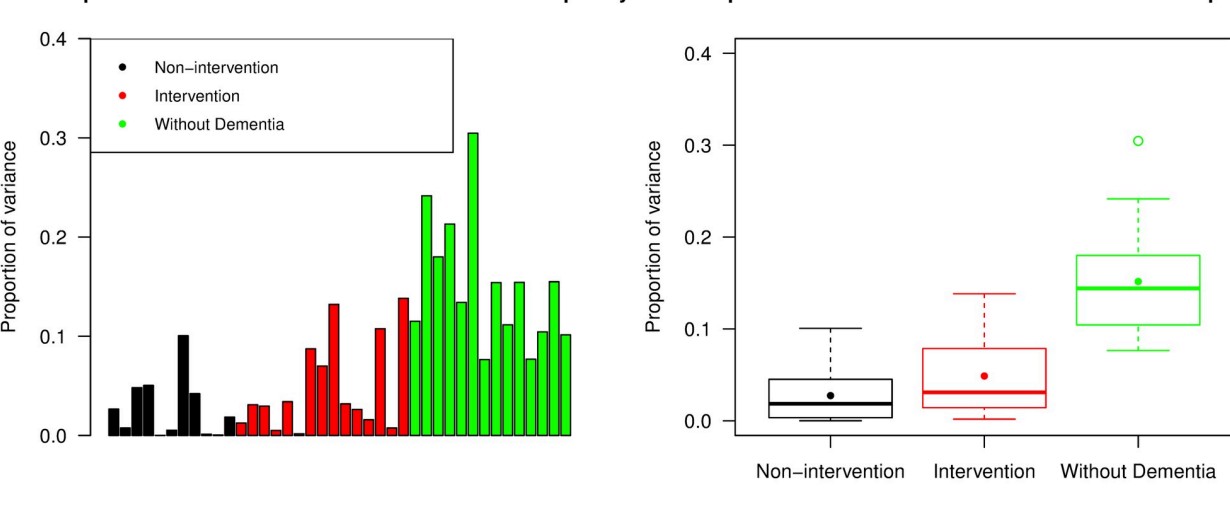

**Fig 10. Proportion of variance around the fundamental frequency (PoV$^{(F)}$).** The left panel displays PoV values explained by the periodogram around the fundamental frequency across all participants, with the same ordering of individuals as the bar chart representing IS values; the right panel is a box plot collating these PoV values across the non-intervention group, the intervention group, and the group of individuals without dementia. For each group, the median and mean of the PoV$^{(F)}$ values are presented by a thick line and a point in its own box, respectively.

participants with advanced dementia. Participants without dementia tended to have higher IS, $\alpha$, PoV$^{(H)}$, but lower IV.

In Table 1, we show the Pearson's correlation coefficients across all participants for all four measures. IV was negatively associated with all other measures, and all other pairwise associations were positively correlated. Such associations across groups were expected given the clear

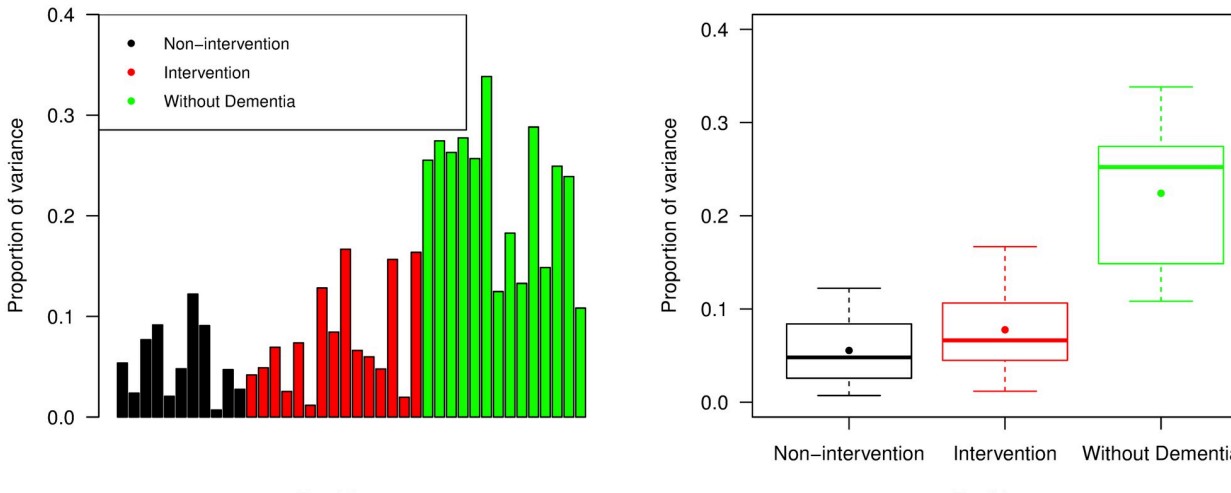

**Fig 11. Proportion of variance around the first four harmonics (PoV$^{(H)}$).** The left panel displays PoV values explained by the periodogram around the first four harmonic frequencies across all participants, with the same ordering of individuals as the bar chart representing IS values; the right panel is a box plot collating these PoV values across the non-intervention group, the intervention group, and the group of individuals without dementia. For each group, the median and mean of the PoV$^{(H)}$ values are presented by a thick line and a point in its own box, respectively.

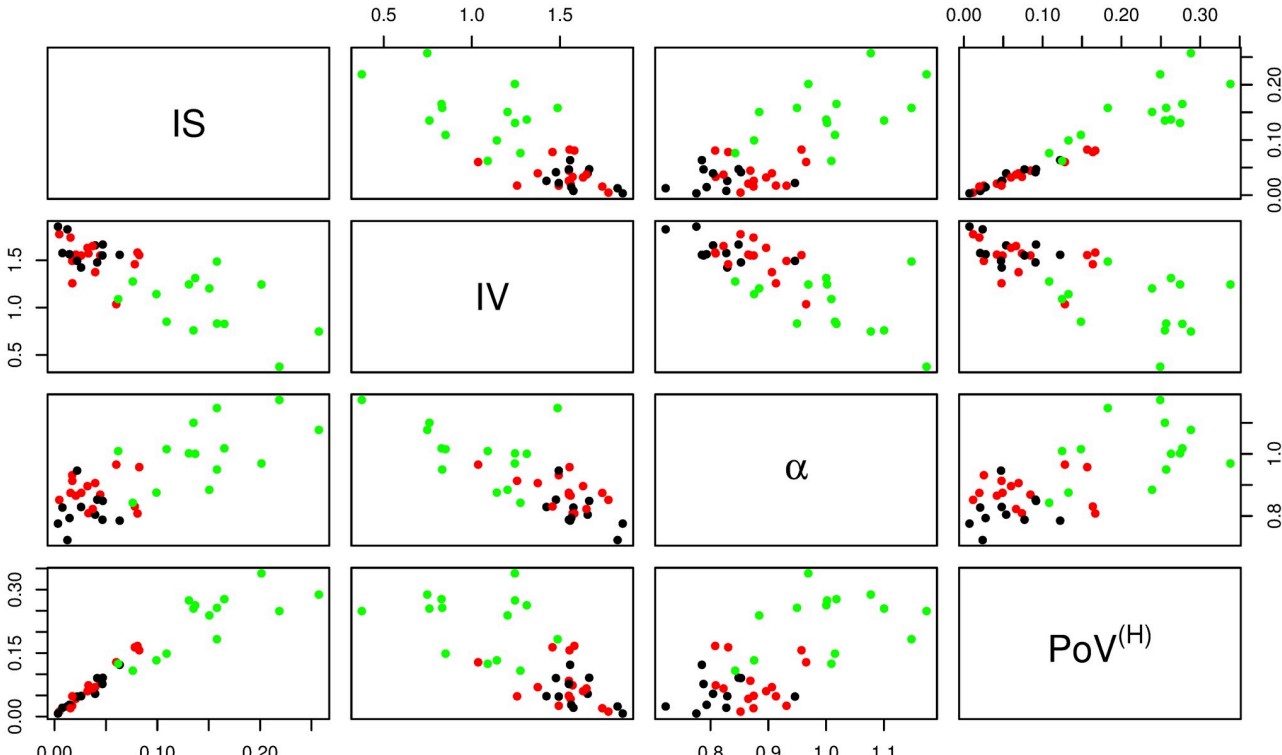

**Fig 12. Scatter plots of all statistical measures.** Scatter plots showing relationships between all possible pairs of the four statistical measures—IS, IV, the DFA scaling exponent ($\alpha$), and PoV$^{(H)}$. Participants from the non-intervention group, the intervention group, and the group of individuals without dementia, are represented by black, red and green dots respectively.

differences between accelerometry time series of participants without dementia and those with advanced dementia. Therefore, to explore relationships between groups, we also separately summarised the Pearson's correlation coefficients for participants with and without advanced dementia in Tables 2 and 3, respectively. From these tables we can see that the relationships between all pairs of measures remained the same within these groups, albeit with lower

**Table 1. Pearson's correlation coefficients between pairs of statistical measures for all participants.**

|  | IS | IV | $\alpha$ | PoV$^{(H)}$ |
|---|---|---|---|---|
| IS |  | -0.775 | 0.735 | 0.943 |
| IV | -0.775 |  | -0.766 | -0.724 |
| $\alpha$ | 0.735 | -0.766 |  | 0.670 |
| PoV$^{(H)}$ | 0.943 | -0.724 | 0.670 |  |

**Table 2. Pearson's correlation coefficients between pairs of statistical measures for participants with advanced dementia (combined non-intervention and intervention groups).**

|  | IS | IV | $\alpha$ | PoV$^{(H)}$ |
|---|---|---|---|---|
| IS |  | -0.348 | 0.148 | 0.983 |
| IV | -0.348 |  | -0.600 | -0.406 |
| $\alpha$ | 0.148 | -0.600 |  | 0.169 |
| PoV$^{(H)}$ | 0.983 | -0.406 | 0.169 |  |

**Table 3. Pearson's correlation coefficients between pairs of statistical measures for participants without dementia.**

|  | IS | IV | $\alpha$ | PoV$^{(H)}$ |
|---|---|---|---|---|
| IS |  | -0.434 | 0.509 | 0.772 |
| IV | -0.434 |  | -0.441 | -0.220 |
| $\alpha$ | 0.509 | -0.441 |  | 0.308 |
| PoV$^{(H)}$ | 0.772 | -0.220 | 0.308 |  |

correlations in general. This demonstrates that these statistical measures have power in separating within group behaviour and are useful statistics that should be considered when performing larger studies, for example to see if certain treatment interventions are effective when circadian rhythms are used as primary outcome.

## Conclusion

This paper proposes four nonparametric summary statistics (IS, IV, the DFA scaling exponent, and PoV) for the study of circadian rhythm and other attributes found in accelerometry data. As a proof-of-concept, we computed each summary statistic on a dataset containing a mixed group of participants, some with advanced dementia, and compared these results with data from individuals without dementia. The analysis shows that these statistics can collectively summarise different features in the data: both the inter-daily features of the circadian cycle (IS and PoV), as well as the intra-daily fragmentation and correlation structure (IV and the DFA scaling exponent). Using these statistics there are clearly different values obtained for participants without dementia and those with advanced dementia, therefore there is a potential for these nonparametric statistics to be used as primary outcomes related with circadian rhythms, or as diagnostic benchmarks and thresholds, which would naturally require more studies and analyses to precisely establish practical clinical guidelines.

Some of the participants with advanced dementia also received a group intervention during the study. As in the original study [32], we did not observe statistically significant differences between the intervention and non-intervention groups. There was however one exception in our paper presented here, where the DFA scaling exponent values (whether daytime, nighttime, or aggregated across both) did show significant difference between non-intervention and intervention groups. Specifically, the scaling exponent was found to be significantly higher for the intervention group which indicated a smoother pattern of behaviour in activity over longer time scales, which was a feature we also found in the group of participants without dementia. Overall, the fact that participants with and without dementia displayed clear differences to each other, but participants within the groups with advanced dementia did not, were both expected results and validated our statistical measures in terms of their efficacy and reliability in summarising key features of accelerometry data. Ultimately to determine the scope for using these summary statistics in a clinical trial setting requires much further study.

The accelerometry data we studied is high frequency—sampled every 5 seconds—and this was typical of modern instruments and studies. In this paper we paid careful consideration as to how statistical methods should be adapted to high frequency sampling. A key finding is that IV should *not* be calculated from hourly subsampling, as has been historically performed in the literature. Hourly subsampling leads to IV values that are unable to effectively separate groups of participants, and have unexpected correlations with other summary statistics. Instead, we proposed subsampling every 5 minutes for our studied dataset, which improved the behaviour and increased the power of this statistic. We note that subsampling at even higher frequencies eventually worsened performance due to contamination from high-

frequency variability. Finally, we note that we proposed a simple aggregation method to ensure no data is lost in the subsampling procedure.

The high frequency nature of the data also allowed us to propose a new metric, *proportion of variance* (PoV), which is a nonparametric spectral-based estimate of circadian strength. As the data is high frequency, there are many frequencies at or near the circadian frequency which can be smoothed over to obtain a stable estimate of circadian strength. A key finding is that the inclusion of *harmonics*, which are integer multiples of the circadian frequency, increased the proportion of variance explained by this statistic and led to improved performance. This statistic is therefore a nonparametric alternative to the parametric cosinor method. We note however that PoV performs similarly to IS with high correlation in these values across participants, and it is therefore possible that only one of these metrics is required, which future studies may reveal.

Finally, the high frequency nature of our data allows for statistics to be calculated to quantify the *memory* or *long-term autocorrelation* structure of the time series. A variety of established techniques exist to quantify such structure, including the DFA method used here. What our analysis revealed however, is the potential for utilising the richness of continuity of accelerometry data to obtain separate daytime and nighttime DFA scaling exponent values, and how this may yield further insight as to the difference between individuals who have disrupted sleep cycles and activity rhythms, and those that do not. In particular, we expect individuals without dementia to have significantly higher DFA scaling exponents at daytime than at nighttime, and this is what our analysis revealed.

This paper we believe is the first to jointly consider three existing nonparametric metrics—IS, IV and the DFA scaling exponent—for analysing accelerometry data, together with proposing a novel nonparametric alternative to cosinor analysis based on aggregating information in the periodogram. We provide guidelines on implementation, and an indication of their performance on a high frequency dataset from people suffering from advanced dementia living in care homes. Limitations of our data analysis include having a relatively small number of participants, and that the without dementia group data comes from individuals from the research group with younger ages than those with advanced dementia living in care homes. Different demographic variables such as age will affect the interpretation of statistical results from accelerometry data and hence, at this point, we cannot conclude that the difference of statistical measures between participants in our case study is solely due to the condition of having advanced dementia or not. Therefore care should be taken in terms of generalising findings from our data analysis, and the performance and robustness of proposed summary statistics should be rechecked in future accelerometry analyses. More broadly, high frequency accelerometry analysis is a relatively recent development in the health sciences, and we very much encourage continued exploration and refinement of statistical methodology as the complexity and dimensionality of acquired datasets continue to grow.

## Supporting information

**S1 Table. Demographic variables.** Key demographic characteristics of the 32 resident participants from the study of [33]. The 26 participants with advanced dementia from which we have valid accelerometry data are all from this larger study, where the demographic breakdown between those participants with and without valid accelerometry data is not available. (PDF)

**S2 Table. Summary statistics.** Mean, standard deviation (SD), minimum and maximum values of all statistical measures used in this paper for different groups of participants. The

dementia group refers to the combined non-intervention and intervention groups.
(PDF)

**S1 Fig. Correlation between IV and IS.** Line plots showing Pearson's correlation coefficients between IV and IS for the ENMO data with subsampling interval for calculating IV varied from 5 seconds to 60 minutes. The non-intervention group, the intervention group, the group of individuals without dementia, and the overall group are presented by black, red, green, and purple lines, respectively. The orange dashed line refers to 5 minutes and indicates our recommended subsampling interval for calculating IV.
(TIF)

**S2 Fig. Correlation between IV and the DFA scaling exponent.** Line plots showing Pearson's correlation coefficients between IV and the DFA scaling exponent for the ENMO data with subsampling interval for calculating IV varied from 5 seconds to 60 minutes. The non-intervention group, the intervention group, the group of individuals without dementia, and the overall group are presented by black, red, green, and purple lines, respectively. The orange dashed line refers to 5 minutes and indicates our recommended subsampling interval for calculating IV.
(TIF)

**S3 Fig. Correlation between IV and PoV$^{(H)}$.** Line plots showing Pearson's correlation coefficients between IV and PoV around the first four harmonic frequencies for the ENMO data with subsampling interval for calculating IV varied from 5 seconds to 60 minutes. The non-intervention group, the intervention group, the group of individuals without dementia, and the overall group are presented by black, red, green, and purple lines, respectively. The orange dashed line refers to 5 minutes and indicates our recommended subsampling interval for calculating IV.
(TIF)

## Author Contributions

**Conceptualization:** Keerati Suibkitwanchai, Adam M. Sykulski, Guillermo Perez Algorta.

**Data curation:** Guillermo Perez Algorta.

**Formal analysis:** Keerati Suibkitwanchai, Daniel Waller.

**Funding acquisition:** Catherine Walshe.

**Investigation:** Keerati Suibkitwanchai.

**Methodology:** Keerati Suibkitwanchai, Daniel Waller.

**Project administration:** Adam M. Sykulski, Guillermo Perez Algorta.

**Resources:** Guillermo Perez Algorta.

**Software:** Keerati Suibkitwanchai, Daniel Waller.

**Supervision:** Adam M. Sykulski, Guillermo Perez Algorta.

**Validation:** Keerati Suibkitwanchai.

**Visualization:** Keerati Suibkitwanchai.

**Writing – original draft:** Keerati Suibkitwanchai.

**Writing – review & editing:** Adam M. Sykulski, Guillermo Perez Algorta, Daniel Waller, Catherine Walshe.

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
