## [Decision Letter · Decision Letter 0]

3 Jun 2020

PONE-D-20-11622

Nonparametric time series summary statistics for high-frequency actigraphy data from individuals with advanced dementia

PLOS ONE

Dear Dr. Suibkitwanchai,

Thank you for submitting your manuscript to PLOS ONE. After careful consideration, I feel that it has merit but does not fully meet PLOS ONE’s publication criteria as it currently stands. Therefore, I invite you to submit a revised version of the manuscript that addresses the points raised during the review process.

I have carefully read the comments of both reviewers and I consider that no one of them have doubts about the contributions of this manuscript. I understand that some technical aspects as well as some results of the experiment need to be clarified, but in my opinion, these are minor issues. Congratulation for the job and I hope to have the revised version soon.

We look forward to receiving your revised manuscript.

Kind regards,

J E. Trinidad Segovia

Academic Editor

PLOS ONE

Journal Requirements:

Reviewers' comments:

Reviewer's Responses to Questions

**Comments to the Author**

1. Is the manuscript technically sound, and do the data support the conclusions?

Reviewer #1: Yes

Reviewer #2: Yes

2. Has the statistical analysis been performed appropriately and rigorously? 

Reviewer #1: Yes

Reviewer #2: Yes

3. Have the authors made all data underlying the findings in their manuscript fully available?

Reviewer #1: Yes

Reviewer #2: Yes

4. Is the manuscript presented in an intelligible fashion and written in standard English?

Reviewer #1: Yes

Reviewer #2: Yes

5. Review Comments to the Author

Reviewer #1: The research problem is to quantify the circadian cycles of populations with and without dementia as a proxy for health. The working hypothesis is that cosinor analysis, which is the standard parametric analysis for circadian rhythms, needs specific cycles of prespecified periods (e.g., 24h) to be present in the data in order to work, and that non-parametric methods are better suited to study this type of data, such as the existing IV, IS and Hurst fractal scaling analysis, and the newly proposed PoV. The authors find that all methods distinguish between the groups with and the group without dementia, and that additionally the Hurst exponent distinguishes between the dementia groups with and without extra treatment.

The research problem is valid, the authors apply all methods adequately, the results obtained are interesting and the paper reads well and is well presented. The focus of the article is more on the statistical analysis than on the physiological interpretation of the numerical results and it would be interesting if the authors could elaborate a little more on the how to interpret the IS, IV, Hurst and PoV values for the different populations. I would recommend publication in PLoS ONE after they solve the following minor problems:

- In clinical research, the name “control group” usually indicates the group of healthy or asymptomatic individuals to be compared to groups of individuals with pathologies. Here, “control group” is used for the individuals with dementia but without additional treatment, compared to a group of individuals with dementia and additional treatment, and a group of individuals without dementia, which is confusing. I understand this name “control group” comes from the previous publications of the authors, refs. [31,32], where only the 2 groups with dementia were studied and that the 3rd group without dementia was added for this article. If the authors do not want to use the name “control group” for the asymptomatic individuals in order to avoid conflict with their previous publications, would it be possible to avoid the name “control” group altogether?

- Are all participants of all 3 groups permanent residents of nursing homes? Are all participants similar with respect to demographic, anthropometric, etc., variables? Could the authors add a table to the main text where group-average values and standard deviations for these variables are compared for all groups, together with p-values to indicate that the only difference between the groups is the condition of dementia and/or the additional treatment?

- I agree with the authors that cosinor might not be the optimal method to analyze circadian cycles in irregular data such as in the case of dementia, but could the authors estimate just how much better non-parametric methods are with respect to a parametric method such as cosinor? I guess that the amplitude of cosinor analysis should be proportional to the power of the fundamental of the spectral analysis, i.e., the measure PoV(F) of the authors? If this is the case, just how much better can the different populations be distinguished by including harmonics as in PoV(H) and/or considering other non-parametric measures such as IS, IV and the Hurst exponent. Can the authors quantitatively confirm that their working hypothesis is correct that the proposed non-parametric measures are better suited to analyze their data? Perhaps in this context it might help to include PoV(F) in Fig. 11 and in the tables 2-4 on p. 13/18?

- I think the data the authors are using should be called “accelerometry” instead of “actigraphy”. Actigraphy is coarse-grained accelerometry according to several different conventions including time above threshold, zero crossings or digital integration (see Ancoli-Israel et al., 2003, Sleep 26: 342), such that the units of actigraphy are usually not the gravity constant g, as in the present article. Ref. [33] which explains the ENMO data used here indeed says, “acceleration signal” and never mentions “actigraphy”. The use of accelerometry instead of actigraphy might actually have helped the authors to realize a separate daytime and night-time analysis with DFA, whereas in the case of actigraphy night-time analysis often becomes difficult because of time series which are mostly zero.

- This is the first time I see an article where Hurst and DFA are treated as synonyms. The two methods are indeed very much related, but I would avoid saying they are the same. The result of DFA analysis is actually the alpha exponent, which varies in a different range than H, with 0<=H<1 and 0<alpha<1.5, 0="" 18="" 6="" analysis="" analysis.="" and="" convert="" exist="" explanation="" formulae="" mathematical="" of="" on="" one="" other="" p.="" that="" the="" to="" values="">

- A related comment, same page p. 6/18, I have never heard about a threshold alpha=1.2, if true then a reference should be added. According to Eke et al (2000), unlike Hurst analysis which must be applied differently to noises (fGn) and to walks (fBm), DFA may be applied to both without distinction. However, some authors have indeed argued that in the cases of walks better results may be obtained without previous integrating the time series before applying DFA (see e.g., Colas et al., 2019, PLoS ONE 14: e0225817), but the explanation here to first differentiate the data (step 5), then to integrate (step 1) and then to add 1 to the slope, seems redundant.

- p. 3/18, the phrase “…the degree of disrupted fractal regulation was strongly negatively correlated with…” is very difficult to understand.

- p. 6/18 how arbitrary are exactly the daytime (06:00-21:00) and night-time (21:00-06:00) intervals? How were these intervals chosen? Do the nursing homes manage fixed lights on and lights out times?

- p. 7/18 If the data is sampled at 1/5 Hz, then the Nyquist frequency should be 2/5 Hz and not 1/10 Hz as stated in the text.

- p. 8/18 and Fig. 1, in the plot of the 24h average of the participant with dementia with additional treatment, why does the circadian cycle seem to be inverted with more activity during the night than during the day?

- p. 9/18 and Fig. 3 on IV as a function of subsampling interval, the curve has a minimum for small subsampling intervals and converges to a constant for large intervals, but this is rather the opposite behavior from your refs. [6] and [9] where a minimum is obtained for intermediate sample intervals and maxima for both small and large intervals. How do you explain this? In particular, how can there be “a dip in intradaily variability (IV) at small sample intervals where there is high-frequency variability in the data” as stated in the text, whereas intuition would suggest a peak?

- Same page, same figure, refs. [6] and [9] suggest to study the whole range of sampling intervals to analyze IV and to interpret physically/physiologically the time scales where differences between populations maximize. What is the physical/physiological meaning of the specific scale of 5min you focus on? Do you think that in other studies, with other populations, other pathologies, and other measurement equipment also a specific scale of 5min will be found, or will this depend on the particular study?

- p. 9/18 and Figs. S1-S3, can you add a legend to explain the colour coding? If the coding is similar to the previous figures, then the meaning of the red, black and green curve is clear, but this is the first time the colour blue is used?

- Same page, same figures. I agree with the argument of the anticorrelation between IV on the one hand and IS or PoV on the other hand. I find the relation with Hurst less clear, why should there be an anticorrelation between IV and Hurst? I do not think Figs. S1-S3 confirm the statement of the specific scale of 5min, because the blue, red and black curves show anticorrelations for all subsampling intervals. Only the green curve passes from anticorrelation to correlation, but at a scale of 30min and not 5min.

- p. 9/18 and Figs. 2, 4 and 6 what is the physical/physiological interpretation of the larger spread in the boxes of IV, IS and DFA exponent for the participants without dementia? I do not see a larger overlap between the boxes of the populations with insomnia for IV than for IS.

- p. 10/18 and Fig. 5, the range of time scales from which the DFA exponent is derived seems rather small, between 2^4=16 and 2^8=256 time units? Are these time units equal to 5s? Then these time scales correspond to 80-1,280s or 1.3-21min, that seems small to characterize daytime duration of 15h or night-time duration of 9h. How do these curves look like at larger time scales, are they linear or might there be crossover effects as in ref. [6]?

- p- 10/18 and Figs. 6&7, I guess that what the authors want to say with “…for individuals without insomnia their data is a combination of stationary and non-stationary time series…”, is that for these individuals their time series behave more as the mysterious 1/f noise, which is indeed the border between stationary noise and non-stationary walks (see Halley et al., 2004; Eke et al., 2000), and which is empirically found in many physiological time series (e.g., heart rate, actigraphy, EEG, etc.) of young and healthy individuals without anyone really understanding why, see e.g. [6], whereas that of the individuals with dementia goes more towards a non-correlated white noise.

- p. 11/18 and Table 1. Why show in the main text a numerical table for the daytime and night-time Hurst exponent and show the numerical results on IS, IV, overall Hurst and PoV(H) in the supplementary information? Perhaps the 2 tables can be joined and showed in the supplementary information?

- p. 11/18 and Fig. 8. The text says that the fundamental and harmonics of the subject with dementia who receives treatment have much smaller powers than the control subject. This is not clear from a first glance of the figure, only when one focusses on the scales of the figures. Could the authors add a 2nd column with the same spectra but where the vertical scale is maintained constant, to facilitate comparison? Also, the units of the powers are not clear, are these percentages or fractions, do in all cases all power sum up to a total of 1 or 100%?

Reviewer #2: In the present article, four nonparametric statistics, i.e., interdaily stability, intradaily variability, Hurst exponent, and the so-called proportion of variance (introduced by the authors) are considered to analyse the circadian rhythm as well as some other features that appear in actigraphy data. In particular, it is worth mentioning that the proportion of variance parameter allows calculating the strength of the circadian rhythm by estimating the spectral density. The authors were also focused on carefully adapting statistical methods to deal with high frequency sampling.

Actigraphy data from a group consisting of 40 subjects were applied the proposed methodology. Specifically, 26 individuals were patients with advanced dementia, whereas the remaining 14 participants did not presented any dementia symptom. All the four summary statistics were calculated for a dataset which contained a mixed group of individuals. The obtained results were compared with respect to a dataset of participants without dementia.

They were obtained several results, all of them being quite interesting.

1.-Overall, thet stated that the four statistics involved in the current article may collectively summarise distinct characteristics of actigraphy data. Thus, interdaily stability and proportion of variance provide information of inter-daily features regarding the circadian cycle. On the other hand, Hurst exponent and intradaily variability throw information about correlation structure and intra-daily fragmentation.

2.-Also, unlike it has been carried out in the literature, they recommend not to calculate intradaily variability by hourly subsampling since it leads to values that do not allow effectively separating groups of individuals and throws unexpected correlations with other statistics. In this way, they propose subsampling every 5 minutes, instead. In additiom they remark that subsampling at higher frequencies eventually worsens the performance due to contamination from high-frequency variability.

3.-They found a high correlation between proportion of variance and interdaily stability parameters, thus suggesting the new parameter as an alternative to interdaily stability.

4.-These statistics were found to behave differently from individuals with dementia to participants without dementia symptoms. As such, that selection of nonparametric statistics may provide a benchmark for further clinical studies.

5.-In particular, they suggested to apply the Hurst exponent separately for daytime and nighttime, resp., thus allowing to separate effects between subjects. In particular, they hypothesised a higher Hurst exponent values at daytime than at nighttime. In this way, their results corroborated that hypothesis.

In my opinion, this paper is very well-written with its content being technically sound. However, let me to provide some comments to be addressed prior to publication.

-In page 2/18, the authors comment on (anti-)persistent processes (resp., series) and Brownian motions, thus providing ranges where their Hurst exponents vary. Anyway, H ranges between 0 and 1. However, at the beginning of page 3/18, they state that for Hurst exponent calculation purposes via the DFA approach, H varies between 0 and 2. That last statement should be corrected in the following terms. If there is a fractal pattern in the series, then the next power law holds: F(n)\\propto n^{\\alpha}, where F(n) denotes the fluctuation for subseries of length equal to n (in the context of the DFA approach) and \\alpha refers to the scaling exponent. That \\alpha corresponds to the slope of a straight line that compares \\log F(n) vs. \\log n. Thus, in the case of nonstationaty time series, we have that \\alpha and the Hurst exponent of the series are connected through the following identity: H=\\alpha(2)-1.

-Though the authors apply the DFA to calculate the Hurst exponent of the series, the authors may comment on some other approaches such as the fractal dimension methods (c.f. Section 3.12 in [MFM19]).

\\bibitem{MFM19}

M.~Fern{\\'{a}}ndez-Mart{\\'{i}}nez, M.A. S{\\'{a}}nchez-Granero, J.E. {Trinidad Segovia}, and Juan~L.G. Guirao, \\emph{{Fractal dimensions for fractal structures with Applications to Finance}}, Springer Nature International Publishing, 2019.

-In page 8/18, they state that both Mann-Whitney test and Student's t-test have been applied to determine whether they are statistically significant differences between the medians/means of the two subsamples. To properly apply the Student's t-test, the subsamples should be normally distributed. Did the authors verify that hypothesis?

-Page 10/18 (and where necessary): please, fix a number of decimal places regarding the p-values that are displayed throughout the text (e.g.: 3 decimal digits) and be consistent with that choice.

-Page 11/18 (and where necessary): please, write the means and standard deviations in the standard format mean\\pm std.

  </alpha<1.5,>

6. PLOS authors have the option to publish the peer review history of their article (what does this mean?). If published, this will include your full peer review and any attached files.

Reviewer #1: No

Reviewer #2: No

---

## [Author Response · Author response to Decision Letter 0]

17 Jul 2020

RESPONSE LETTER

Editor’s comments: I have carefully read the comments of both reviewers and I consider that no one of them have doubts about the contributions of this manuscript. I understand that some technical aspects as well as some results of the experiment need to be clarified, but in my opinion, these are minor issues. Congratulation for the job and I hope to have the revised version soon.

Response: Thank you for these positive comments and inviting us to submit a revised version of our manuscript. We believe we have addressed all the interesting questions and concerns the reviewers have raised, and we thank them for their expertise and insights. We provide our responses in the usual point-by-point format below, where we highlight the changes we have made in the manuscript to reflect their comments.

Reviewer #1

Comment #1: In clinical research, the name “control group” usually indicates the group of healthy or asymptomatic individuals to be compared to groups of individuals with pathologies. Here, “control group” is used for the individuals with dementia but without additional treatment, compared to a group of individuals with dementia and additional treatment, and a group of individuals without dementia, which is confusing. I understand this name “control group” comes from the previous publications of the authors, refs. [31,32], where only the 2 groups with dementia were studied and that the 3rd group without dementia was added for this article. If the authors do not want to use the name “control group” for the asymptomatic individuals in order to avoid conflict with their previous publications, would it be possible to avoid the name “control” group altogether?

Response: Yes, we agree with your suggestion. We have changed the name of our groups of participants to (1) non-intervention group, (2) intervention group, and (3) group without dementia. 

Comment #2: Are all participants of all 3 groups permanent residents of nursing homes? Are all participants similar with respect to demographic, anthropometric, etc., variables? Could the authors add a table to the main text where group-average values and standard deviations for these variables are compared for all groups, together with p-values to indicate that the only difference between the groups is the condition of dementia and/or the additional treatment?

Response: Thank you for pointing out that the details of the study participants should be made clearer here. We have revised the beginning of the Materials and Methods section to make clearer the differences between the groups. In particular, we make clear that all participants with advanced dementia resided in care homes. However, participants that provided data for the group without dementia were members of the research team or academic colleagues that collaborated with the project during the phase of piloting protocol as described in the Materials and Methods section. Following your suggestion, we have added a table to the Supporting Information (Table S1) with demographic information for care home participants adapted from Froggatt et al., 2018 [33]. For this paper, we have analysed data from those participants with valid accelerometry information (26 in total). For participants in the group without dementia, we do not have demographic information available. We have also included discussion of the limitations of our data analysis in the final paragraph of the conclusions section at the top of Page 20 of the revised manuscript.

Comment #3: I agree with the authors that cosinor might not be the optimal method to analyze circadian cycles in irregular data such as in the case of dementia, but could the authors estimate just how much better non-parametric methods are with respect to a parametric method such as cosinor? I guess that the amplitude of cosinor analysis should be proportional to the power of the fundamental of the spectral analysis, i.e., the measure PoV(F) of the authors? If this is the case, just how much better can the different populations be distinguished by including harmonics as in PoV(H) and/or considering other non-parametric measures such as IS, IV and the Hurst exponent. Can the authors quantitatively confirm that their working hypothesis is correct that the proposed non-parametric measures are better suited to analyze their data?

Response: Thank you for raising this interesting point. Indeed, we have applied cosinor analysis to our accelerometry data and compared to our nonparametric measures. We found that the correlation between the amplitude from the cosinor model and PoV at the fundamental frequency, or PoV(F), is about 0.677. Furthermore, we found the correlation between the coefficient of determination (R2) by fitting this model and PoV(F) to be 0.977. These results indicate a strong linear relationship between these two methods as per your intuition. However, the coefficient of determination (R2) by fitting the cosinor model is quite poor (less than 0.1) and there is high variability of R2 within each group of participants. This is consistent with findings from Fossion et al., 2017 [6], where again the cosinor model is considered a poor fit to the data, especially from individuals with dementia. The difference between cosinor and our proposed method becomes more apparent when we include harmonics. The correlation between R2 from the cosinor model, and PoV around the first four harmonics or PoV(H), drops to 0.901. Moreover, the PoV(H) value has an average of 0.224 for participants without dementia, indicating that we can explain 22.4% of the variability in the time series with this simple measure. In contrast the average coefficient of determination among participants without dementia from cosinor is only 0.079, thus only explaining 7.9% of the variability. A discussion of these findings can be found in the Results and Discussion section at the end of the PoV section (bottom of page 16 and top of page 17). 

Comment #4: I think the data the authors are using should be called “accelerometry” instead of “actigraphy”. Actigraphy is coarse-grained accelerometry according to several different conventions including time above threshold, zero crossings or digital integration (see Ancoli-Israel et al., 2003, Sleep 26: 342), such that the units of actigraphy are usually not the gravity constant g, as in the present article. Ref. [33] which explains the ENMO data used here indeed says, “acceleration signal” and never mentions “actigraphy”. The use of accelerometry instead of actigraphy might actually have helped the authors to realize a separate daytime and night-time analysis with DFA, whereas in the case of actigraphy night-time analysis often becomes difficult because of time series which are mostly zero. 

Response: We totally agree with your suggestion. As such, we have changed the word “actigraphy” to “accelerometry” in all places in the manuscript. Our new title now becomes “Nonparametric time series summary statistics for high-frequency accelerometry data from individuals with advanced dementia.”

Comment #5: This is the first time I see an article where Hurst and DFA are treated as synonyms. The two methods are indeed very much related, but I would avoid saying they are the same. The result of DFA analysis is actually the alpha exponent, which varies in a different range than H, with 0<=H<1 and 0<alpha<1.5, and mathematical formulae exist to convert the values of one analysis to values of the other analysis. The explanation on p. 6/18 that 0<H<1 (stationary noise) and 1<H2 (non-stationary walk) seems wrong, the range is always 0<=H<1, both for noises and for walks but with different interpretation, see e.g., Eke et al, 2000, Eur. J. Physiol. 439: 403 and Halley et al., 2004, Fluct Noise Lett. 4: R1. 

Response: We completely agree that Hurst and DFA are related, but not exactly the same as you have described. As such, we have replaced reference to the “Hurst exponent (H)” with the “DFA scaling exponent (α)”, which is estimated by the slope of the linear regression line from the DFA method. The value of α is between zero and two such that α = H for time series with stationary noise-like behaviour and α = H + 1 for time series with nonstationary random walk-like behaviour. We have rewritten the relevant part of the introduction (found on pages 2-3) and the DFA scaling exponent subsection in the Materials and Methods section (found on page 6) to clarify this distinction. 

Comment #6: A related comment, same page p. 6/18, I have never heard about a threshold alpha=1.2, if true then a reference should be added. According to Eke et al (2000), unlike Hurst analysis which must be applied differently to noises (fGn) and to walks (fBm), DFA may be applied to both without distinction. However, some authors have indeed argued that in the cases of walks better results may be obtained without previous integrating the time series before applying DFA (see e.g., Colas et al., 2019, PLoS ONE 14: e0225817), but the explanation here to first differentiate the data (step 5), then to integrate (step 1) and then to add 1 to the slope, seems redundant.

Response: Thank you for your explanation with these useful references. The threshold of 1.2 comes from Ihlen, 2012 [21], however this is suggested to provide better results in the estimation of the q-order scaling exponent in multifractal DFA. In our case, as we performed monofroctal DFA (and found no evidence to extend to multifractal DFA), then we agree this step is redundant. As such, we have removed it from our procedure described on Page 6, thank you for spotting this. 

Comment #7: p. 3/18, the phrase “…the degree of disrupted fractal regulation was strongly negatively correlated with…” is very difficult to understand.

Response: We apologise for this confusing sentence. To make it clearer, we have revised this phrase to “the change of DFA scaling exponent could be found from ante-mortem actigraphy records of some patients with dementia and its degree of change was negatively correlated with the number of two major circadian neurotransmitters found in the suprachiasmatic nucleus.”

Comment #8: p. 6/18 how arbitrary are exactly the daytime (06:00-21:00) and night-time (21:00-06:00) intervals? How were these intervals chosen? Do the nursing homes manage fixed lights on and lights out times?

Response: Thank you for seeking clarification here. Because each participant had a different sleeping period, and there was no available data from the care homes regarding daily schedules, then setting fixed daytime and nighttime periods a priori was challenging. Instead we chose these intervals using a data-driven approach by analysing average sleeping patterns as implied from average ENMO readings. We include a new figure in the revised manuscript (Figure 7 on page 13) which shows average ENMO readings across groups and all participants. Based on the findings of this figure, we have actually revised our daytime and nighttime intervals to 06:00-23:00 and 23:00-06:00, respectively. We note that this has not significantly changed the results from our findings in Figure 8 that follows, nor the acceptances/rejections from the p-value tests reported. In the revised manuscript, the text which discusses the choice of windows for daytime and nighttime analysis can be found at the bottom of Page 12.

Comment #9: p. 7/18 If the data is sampled at 1/5 Hz, then the Nyquist frequency should be 2/5 Hz and not 1/10 Hz as stated in the text. 

Response: The Nyquist frequency, which is not the same as the Nyquist rate, is equal to half of the sampling rate. From this definition, the Nyquist frequency of 1/10 Hz is correct. However, we have edited the wording in our paper to make this clearer (the first paragraph on page 7).

Comment #10: p. 8/18 and Fig. 1, in the plot of the 24h average of the participant with dementia with additional treatment, why does the circadian cycle seem to be inverted with more activity during the night than during the day?

Response: This is an interesting question. The reason why we presented this plot is to show the irregular pattern that can be found from some participants especially those with advanced dementia. This high variability at night is quite interesting and something we attempt to capture in our statistics (PoV, nighttime DFA etc). However, as this is not fully representative of a typical participant, we have changed this to a time-averaged plot from another participant in the same group. The new plot has slightly more fluctuation during daytime than nighttime, as expected, and this pattern can also be found in the new figure (Figure 7) of 24-hour ENMO averaging over all participants within each group of individuals.

Comment #11: p. 9/18 and Fig. 3 on IV as a function of subsampling interval, the curve has a minimum for small subsampling intervals and converges to a constant for large intervals, but this is rather the opposite behavior from your refs. [6] and [9] where a minimum is obtained for intermediate sample intervals and maxima for both small and large intervals. How do you explain this? In particular, how can there be “a dip in intradaily variability (IV) at small sample intervals where there is high-frequency variability in the data” as stated in the text, whereas intuition would suggest a peak?

Response: Thank you for raising this interesting point. We have rechecked these two references and found that our pattern of IV as a function of subsampling interval is in fact consistent with several results from ref. [9] (see Figs 2, 3 and 5 of this paper). However, other relationships such as the “U-shape” that you describe have also been found in this reference and ref. [6]. Mathematically both relationships are possible, and this will depend on the autocorrelation characteristics of the data being studied, as well as possibly the recording device used. In our case, the initial rapid rise in IV as the subsampling interval from 5 seconds to higher multiple values was due to the large positive autocorrelations in the time series at these small lags. In the revised manuscript we include a comparison with refs. [6] and [9] on Page 9. Then on Page 11 we recommend that such analyses be repeated in future accelerometry studies given the best choice of subsampling appears to be dependent on the study being carried out. Furthermore, given the study-specific nature of our findings, we have made clear that our recommendation of 5 minutes is specific to this study (both in the abstract and conclusion) and we have also suggested a range of 2 to 10 minutes which broadly achieves the same desirable results of the IV statistic. 

Comment #12: Same page, same figure, refs. [6] and [9] suggest to study the whole range of sampling intervals to analyze IV and to interpret physically/physiologically the time scales where differences between populations maximize. What is the physical/physiological meaning of the specific scale of 5min you focus on? Do you think that in other studies, with other populations, other pathologies, and other measurement equipment also a specific scale of 5min will be found, or will this depend on the particular study?

Response: This is a very interesting question. As we discussed in our response to your previous comment, we believe the appropriate subsampling rate for IV is study-specific, and as detailed we have made such commentary in the revised paper. We also studied the whole range of subsampling intervals to find our best subsampling interval. In the paper, we reported results from the highest frequency (every 5 seconds) to one-hourly subsampling. We found that performance of the IV statistic drops rapidly by subsampling less frequently than hourly so we did not report findings beyond this upper limit. The choice of 5 minutes for our study (and the broader recommended range of 2 to 10 minutes) is based on the evidence of the data and not any physiological reason, although we do agree that the latter would be an interesting subject of further study but is beyond the scope of this paper. Instead we have focused on an evidence-based approach from the data, where we could really benefit from the large volumes of data we have due to the high frequency and lengthy recordings available amounting to the order of 50,000 datapoints per participant.

Comment #13: p. 9/18 and Figs. S1-S3, can you add a legend to explain the colour coding? If the coding is similar to the previous figures, then the meaning of the red, black and green curve is clear, but this is the first time the colour blue is used?

Response: We have added legends to these supplementary figures for better visualisation.

Comment #14: Same page, same figures. I agree with the argument of the anticorrelation between IV on the one hand and IS or PoV on the other hand. I find the relation with Hurst less clear, why should there be an anticorrelation between IV and Hurst? I do not think Figs. S1-S3 confirm the statement of the specific scale of 5min, because the blue, red and black curves show anticorrelations for all subsampling intervals. Only the green curve passes from anticorrelation to correlation, but at a scale of 30min and not 5min.

Response: Thank you for your comment. We have revised the text to better explain the relationship between IV and the DFA scaling exponent. We have added the sentence “High IV and low DFA exponents are synonymous with rougher more volatile time series, whereas low IV and high DFA exponents correspond to smoother more meandering time series.” This means that the anticorrelation between these two metrics from accelerometry data is expected. The below plot shows an example of this anticorrelation derived from a fractional Gaussian noise (fGn) process, where the Hurst exponent (H) is the same as the DFA scaling exponent (α). Here we have simulated fGns across a range of H values and estimated the IV statistic from these time series. 

Regarding the effect of the subsampling interval, indeed the supplementary plots show anticorrelations for subsampling intervals less than 20 minutes or so in all cases of study. However, we found that the range of subsampling intervals for which particularly strong negative correlations are found is from 2 to 10 minutes. The specific time scale of 5 minutes was chosen although we find that in practice all values in this range can be used. We have revised our discussion and interpretation of these supplementary figures in the text at the end of page 10 and beginning of page 11 to reflect these points. 

Fig 1. The relationship between the Hurst exponent (H) (which is here the same as the DFA scaling exponent or α) used to explain the fractal behaviour of the fractional Gaussian noise process and the corresponding IV calculated by Eqs (2) - (4) in the manuscript

Comment #15: p. 9/18 and Figs. 2, 4 and 6 what is the physical/physiological interpretation of the larger spread in the boxes of IV, IS and DFA exponent for the participants without dementia? I do not see a larger overlap between the boxes of the populations with insomnia for IV than for IS. 

Response: Thank you for your question. We suspect that is because the participants without dementia did not have similar living conditions to each other, unlike the participants with advanced dementia residing in nursing homes, then this contributed to the high variability of some of the metrics. For the comparison of overlapping between boxes, we agree with your comment that a larger overlap is not clearly observed, so we have removed this sentence from the manuscript.

Comment #16: p. 10/18 and Fig. 5, the range of time scales from which the DFA exponent is derived seems rather small, between 2^4=16 and 2^8=256 time units? Are these time units equal to 5s? Then these time scales correspond to 80-1,280s or 1.3-21min, that seems small to characterize daytime duration of 15h or night-time duration of 9h. How do these curves look like at larger time scales, are they linear or might there be crossover effects as in ref. [6]?

Response: This is an interesting suggestion about the time scales for DFA analysis. We extended the range of time scales to be between 24 = 16 and 212 = 4096 samples for three participants, one from each group. Since our sampling period is 5 seconds, this range is therefore between 1.333 min and 341.333 min. From the below plot, the DFA scaling exponent is estimated by the slope of linear regression line for each participant. Because there is no change of slope in each line, the single estimated value of the DFA scaling exponent is enough to characterise the fractal behaviour in the whole range of the time series. However, we observed that using this larger range of time scales is more computationally expensive without significantly changing the estimated exponent across all participants. As a result, we prefer using the existing range of time scales for our data analysis in the paper (2^4 to 2^8) such that the code runs efficiently. We have commented on this in the paper on Page 6 in the second step of the DFA algorithm. 

Fig 2. Data points and their linear regression line for the estimation of DFA scaling exponents (slopes) from three participants

Comment #17: p- 10/18 and Figs. 6&7, I guess that what the authors want to say with “…for individuals without insomnia their data is a combination of stationary and non-stationary time series…”, is that for these individuals their time series behave more as the mysterious 1/f noise, which is indeed the border between stationary noise and non-stationary walks (see Halley et al., 2004; Eke et al., 2000), and which is empirically found in many physiological time series (e.g., heart rate, actigraphy, EEG, etc.) of young and healthy individuals without anyone really understanding why, see e.g. [6], whereas that of the individuals with dementia goes more towards a non-correlated white noise. 

Response: This is a very good suggestion. We have revised our sentence here so that a time series with DFA scaling exponent close to 1 is referred to as pink noise or 1/f noise, where this is the boundary between a stationary noise and a nonstationary random walk. This revised text can now be found on Page 12.

Comment #18: p. 11/18 and Table 1. Why show in the main text a numerical table for the daytime and night-time Hurst exponent and show the numerical results on IS, IV, overall Hurst and PoV(H) in the supplementary information? Perhaps the 2 tables can be joined and showed in the supplementary information?

Response: Yes, we agree with your comment. We have now merged these two tables together and presented in the supplementary information as one table (S2 Table). 

Comment #19: p. 11/18 and Fig. 8. The text says that the fundamental and harmonics of the subject with dementia who receives treatment have much smaller powers than the control subject. This is not clear from a first glance of the figure, only when one focusses on the scales of the figures. Could the authors add a 2nd column with the same spectra but where the vertical scale is maintained constant, to facilitate comparison? Also, the units of the powers are not clear, are these percentages or fractions, do in all cases all power sum up to a total of 1 or 100%?

Response: Thank you for this suggestion. First, the unit of the power spectral density as estimated by the periodogram is g2/Hz, and we have marked this in the revised figure. To facilitate comparison between periodograms in this figure, which is now Fig. 9, we have included three more plots showing the values of the periodogram at each frequency as a percentage of the total variance (which is equivalently the area of the periodogram). These plots have the same vertical range between 0% and 10% to facilitate comparison. The participant without dementia has higher percentage values in the periodogram at the fundamental and harmonic frequencies than those with advanced dementia, as expected.

Reviewer #2

Comment #20: In page 2/18, the authors comment on (anti-)persistent processes (resp., series) and Brownian motions, thus providing ranges where their Hurst exponents vary. Anyway, H ranges between 0 and 1. However, at the beginning of page 3/18, they state that for Hurst exponent calculation purposes via the DFA approach, H varies between 0 and 2. That last statement should be corrected in the following terms. If there is a fractal pattern in the series, then the next power law holds: F(n)\propto n^{\alpha}, where F(n) denotes the fluctuation for subseries of length equal to n (in the context of the DFA approach) and \alpha refers to the scaling exponent. That \alpha corresponds to the slope of a straight line that compares \log F(n) vs. \log n. Thus, in the case of nonstationaty time series, we have that \alpha and the Hurst exponent of the series are connected through the following identity: H=\alpha(2)-1.

Response: This is a very nice suggestion, and we apologise for our lack of clarity. As such, we have replaced reference to the “Hurst exponent (H)” with the “DFA scaling exponent (α)”, which is estimated by the slope of the linear regression line from the DFA method. The value of α (not H) is between 0 and 2 such that α = H for time series with stationary noise-like behaviour and α = H + 1 for time series with nonstationary random walk-like behaviour, and H is between 0 and 1 always as you state. This revised text can be found both on the bottom of Page 2 in the revised introduction, and on Page 6 where the DFA method is formally introduced.

Comment #21: Though the authors apply the DFA to calculate the Hurst exponent of the series, the authors may comment on some other approaches such as the fractal dimension methods (c.f. Section 3.12 in [MFM19]).

\bibitem{MFM19}

 M.~Fern{\'{a}}ndez-Mart{\'{i}}nez, M.A. S{\'{a}}nchez-Granero, J.E. {Trinidad Segovia}, and Juan~L.G. Guirao, \emph{{Fractal dimensions for fractal structures with Applications to Finance}}, Springer Nature International Publishing, 2019.

Response: Thank you for your suggestion with this reference. There are indeed a variety of techniques to quantify the fractal structure including this fractal dimension (FD) method. As such, we have added a sentence reflecting this, and included this reference, in the revised introduction (see revised text at top of Page 3).

Comment #22: In page 8/18, they state that both Mann-Whitney test and Student's t-test have been applied to determine whether they are statistically significant differences between the medians/means of the two subsamples. To properly apply the Student's t-test, the subsamples should be normally distributed. Did the authors verify that hypothesis?

Response: Thank you for your question. We applied the Mann-Whitney U test to our data because we did not wish to enforce a specific distribution on the statistics. From our analysis, some of these statistics are indeed not normally distributed, so the Student’s t-test would not be reliable here. The reason why we mentioned t-tests is to show that they provide the same results in terms of significance as the Mann-Whitney U test, even though the assumption of normality is violated. However, we agree that this information adds little and is perhaps confusing, so we have deleted any reference to Student’s t-tests to avoid any confusion, and all our reported p-values are exclusively from the Mann-Whitney U test.

Comment #23: Page 10/18 (and where necessary): please, fix a number of decimal places regarding the p-values that are displayed throughout the text (e.g.: 3 decimal digits) and be consistent with that choice.

Response: We have fixed all decimal numbers in the manuscript to be displayed with three decimal digits. 

Comment #24: Page 11/18 (and where necessary): please, write the means and standard deviations in the standard format mean\pm std.

Response: Thank you for pointing this out. The representation of means and standard deviations has been changed to its standard format (mean±sd).

Other changes

We have changed the normalisation of our Proportion of Variance (PoV) method. This now takes both positive and negative fundamental frequencies along with their harmonics into account for the PoV calculation. Due to the symmetric property of the power spectrum, the new PoV values will be double the previous values. However, there is no change in the interpretation of PoV results and the statistically significant relationships between PoV and the other metrics from these findings (i.e. the p-values are all unchanged). In response to reviewer comments, we have also added a new figure (Fig. 7) presenting average ENMO values over all participants in each group and all groups together for the selection of our new appropriate daytime and nighttime periods and a new reference (ref. 23) for an alternative DFA exponent estimation method. These will result in changes to the numbering of some figures and references.

---

## [Decision Letter · Decision Letter 1]

12 Aug 2020

PONE-D-20-11622R1

Nonparametric time series summary statistics for high-frequency accelerometry data from individuals with advanced dementia

PLOS ONE

Dear Dr. Suibkitwanchai,

Thank you for submitting your manuscript to PLOS ONE. After careful consideration, I can see that major concerns have been properly addresses by authors. However, one of the reviewers still have some doubts that need to be answered. Therefore, I invite you to submit a revised version of the manuscript that addresses the points raised during the review process.

We look forward to receiving your revised manuscript.

Kind regards,

J E. Trinidad Segovia

Academic Editor

PLOS ONE

Reviewers' comments:

Reviewer's Responses to Questions

**Comments to the Author**

1. If the authors have adequately addressed your comments raised in a previous round of review and you feel that this manuscript is now acceptable for publication, you may indicate that here to bypass the “Comments to the Author” section, enter your conflict of interest statement in the “Confidential to Editor” section, and submit your "Accept" recommendation.

Reviewer #1: (No Response)

Reviewer #2: All comments have been addressed

2. Is the manuscript technically sound, and do the data support the conclusions?

Reviewer #1: Yes

Reviewer #2: Yes

3. Has the statistical analysis been performed appropriately and rigorously? 

Reviewer #1: Yes

Reviewer #2: Yes

4. Have the authors made all data underlying the findings in their manuscript fully available?

Reviewer #1: Yes

Reviewer #2: Yes

5. Is the manuscript presented in an intelligible fashion and written in standard English?

Reviewer #1: Yes

Reviewer #2: Yes

6. Review Comments to the Author

Reviewer #1: The authors have answered satisfactorily to most of my previous comments, but I still have 3 comments left. The first comment should be taken into account by the authors because it is relevant for the conclusions they draw from the study, whereas I leave the the other two comments to consideration of the authors.

1) The study compares 3 populations: i) non-intervention group and ii) intervention group, for which demographic data is available in S1_Table, and iii) group without dementia that corresponds to members of the research team and colleagues, and for which no demographic data is available. Groups i) and ii) are elderly adults with an average age of 79-85yo, whereas group iii) are active young or mature adults. Is it correct to compare groups i) and ii) on the one hand and group iii) on the other hand? If so, what does this comparison teach us? Clearly differences are not only about the presence or absence of dementia. The conclusion section should be redacted accordingly.

2) The manuscript refers both to "actigraphy" and "accelerometry". The data used in the present study was accelerometry, whereas most of the previous studies on which the present study builds used actigraphy. It might be useful to explain in the introduction the difference between both types of data, which may not be clear for all readers.

3) Do the authors think that the "U" shape of IV as a function of the sampling frequency in refs. [6,9] and the different shape of their IV curve may be due to the fact that refs. [6.9] were using actigraphy data whereas in the present manuscript accelerometry data was used? Clearly both types of data will have different kinds of autocorrelation properties.

Reviewer #2: The authors have properly dealt with all the issues I pointed out in my previous revision. As such, in my opinion, this article is ready to be accepted for publication in Plos One.

7. PLOS authors have the option to publish the peer review history of their article (what does this mean?). If published, this will include your full peer review and any attached files.

Reviewer #1: No

Reviewer #2: **Yes: **M. Fernández-Martínez

---

## [Author Response · Author response to Decision Letter 1]

28 Aug 2020

Thank you for your comments and inviting us to submit a revised version of our manuscript. We would like to thank both reviewers for their expertise and insights into our research. We believe we have addressed all three interesting questions and concerns the first reviewer has raised. We provide our responses in the usual point-by-point format below, where we highlight the changes we have made in the manuscript to reflect their comments.

Reviewer #1

Comment #1: The study compares 3 populations: i) non-intervention group and ii) intervention group, for which demographic data is available in S1_Table, and iii) group without dementia that corresponds to members of the research team and colleagues, and for which no demographic data is available. Groups i) and ii) are elderly adults with an average age of 79-85yo, whereas group iii) are active young or mature adults. Is it correct to compare groups i) and ii) on the one hand and group iii) on the other hand? If so, what does this comparison teach us? Clearly differences are not only about the presence or absence of dementia. The conclusion section should be redacted accordingly.

Response: Thank you for pointing this out, and we agree that the precise wording of the conclusions required some modification. Please find the new wording in the Conclusion of the revised manuscript (page 16 now reads “Limitations of our data analysis include having a relatively small number of participants, and that the without dementia group data comes from individuals from the research group with younger ages...”), where we have made clear the limitation that the without dementia group has a different demographic profile, meaning that we cannot conclude that the difference of statistical measures between participants is solely due to the condition of have advanced dementia or not.

Comment #2: The manuscript refers both to "actigraphy" and "accelerometry". The data used in the present study was accelerometry, whereas most of the previous studies on which the present study builds used actigraphy. It might be useful to explain in the introduction the difference between both types of data, which may not be clear for all readers.

Response: Thank you again for pointing out this important distinction. We have now been consistent and clear on this point in the revised manuscript. As such we have modified the first few lines of the Introduction (Page 1 now reads “Accelerometry data provides a method for monitoring physical activity over time...”) to make clear that we work with accelerometry data, and then in the Materials and Methods Section we make clear the precise difference between ENMO/accelerometry data and actigraphy data such as counts per time interval (Page 4 now reads “We emphasise that in the literature other forms of actigraphy time series are sometimes studied...”).

Comment #3: Do the authors think that the "U" shape of IV as a function of the sampling frequency in refs. [6,9] and the different shape of their IV curve may be due to the fact that refs. [6.9] were using actigraphy data whereas in the present manuscript accelerometry data was used? Clearly both types of data will have different kinds of autocorrelation properties.

Response: We agree that the differences here may be due to the difference in type of accelerometry/actigraphy data studied. As such we have acknowledged this possibility in the discussion (Page 9 now reads “We note however that these studies were on a range of simulated data, as well as human and animal participants, and used different actigraphy output by studying the counts per time interval rather than ENMO accelerometry data...”). We hope that this analysis is repeated in future experiments to shed further light on the interesting question of optimal subsampling for calculating IV.

Thank you once again for sharing your expertise and your thorough reading of the paper, which we feel that our manuscript has significantly been improved after incorporating your suggested revisions.

---

## [Decision Letter · Decision Letter 2]

7 Sep 2020

Nonparametric time series summary statistics for high-frequency accelerometry data from individuals with advanced dementia

PONE-D-20-11622R2

Dear Dr. Suibkitwanchai,

We’re pleased to inform you that your manuscript has been judged scientifically suitable for publication and will be formally accepted for publication once it meets all outstanding technical requirements.

Kind regards,

J E. Trinidad Segovia

Academic Editor

PLOS ONE

Additional Editor Comments (optional):

Reviewers' comments:

Reviewer's Responses to Questions

**Comments to the Author**

1. If the authors have adequately addressed your comments raised in a previous round of review and you feel that this manuscript is now acceptable for publication, you may indicate that here to bypass the “Comments to the Author” section, enter your conflict of interest statement in the “Confidential to Editor” section, and submit your "Accept" recommendation.

Reviewer #1: All comments have been addressed

2. Is the manuscript technically sound, and do the data support the conclusions?

Reviewer #1: Yes

3. Has the statistical analysis been performed appropriately and rigorously? 

Reviewer #1: Yes

4. Have the authors made all data underlying the findings in their manuscript fully available?

Reviewer #1: Yes

5. Is the manuscript presented in an intelligible fashion and written in standard English?

Reviewer #1: Yes

6. Review Comments to the Author

Reviewer #1: (No Response)

7. PLOS authors have the option to publish the peer review history of their article (what does this mean?). If published, this will include your full peer review and any attached files.

Reviewer #1: No

---

## [Editor Report · Acceptance letter]

16 Sep 2020

PONE-D-20-11622R2 

Nonparametric time series summary statistics for high-frequency accelerometry data from individuals with advanced dementia 

Dear Dr. Suibkitwanchai:

I'm pleased to inform you that your manuscript has been deemed suitable for publication in PLOS ONE. Congratulations! Your manuscript is now with our production department. 

Kind regards, 

on behalf of

Dr. J E. Trinidad Segovia 

Academic Editor

PLOS ONE